# ZERO-SHOT FORECASTING OF CHAOTIC SYSTEMS

**Yuanzhao Zhang**
Santa Fe Institute
Santa Fe, NM, USA

**William Gilpin**$^*$
Department of Physics
University of Texas at Austin
Austin, TX, USA

## ABSTRACT

Time-series forecasting is a challenging problem that traditionally requires specialized models custom-trained for the specific task at hand. Recently, inspired by the success of large language models, foundation models pre-trained on vast amounts of time-series data from diverse domains have emerged as a promising candidate for general-purpose time-series forecasting. The defining characteristic of these foundation models is their ability to perform zero-shot learning, that is, forecasting a new system from limited context data without explicit re-training or fine-tuning. Here, we evaluate whether the zero-shot learning paradigm extends to the challenging task of forecasting chaotic systems. Across 135 distinct chaotic dynamical systems and $10^8$ timepoints, we find that foundation models produce competitive forecasts compared to custom-trained models (including NBEATS, TiDE, etc.), particularly when training data is limited. Interestingly, even after point forecasts fail, large foundation models are able to preserve the geometric and statistical properties of the chaotic attractors. We attribute this success to foundation models' ability to perform in-context learning and identify context parroting as a simple mechanism used by these models to capture the long-term behavior of chaotic dynamical systems. Our results highlight the potential of foundation models as a tool for probing nonlinear and complex systems.

## 1 INTRODUCTION

Classical paradigms in machine learning (ML) require the model to be trained on data specific to the intended task. For example, to forecast the weather in Singapore, a model would need to be trained on past weather data from Singapore. However, recent work in statistical learning has highlighted the power of generative pre-trained models, which use probabilistic approaches and vast amounts of training data to build foundation models that can excel at diverse tasks without the need for separate retraining. In time-series forecasting, this paradigm shift has ignited an intense race to build general-purpose pre-trained models that can make zero-shot forecasts for any time series (Oreshkin et al., 2021; Garza & Mergenthaler-Canseco, 2023; Rasul et al., 2023; Jin et al., 2023; Gruver et al., 2024; Dooley et al., 2024; Liu et al., 2024b; Woo et al., 2024; Ansari et al., 2024; Goswami et al., 2024). Such models have seen some initial success in forecasting real-world time series (Liang et al., 2024) but they have not been systematically tested on chaotic dynamical systems, especially in terms of their performance in long-term forecasting over an extended time horizon.

There are several reasons why such tests are interesting. First, to train foundation models for time series, the amount of high-quality time-series data needed is the single most crucial bottleneck. For this reason, a significant percentage of openly-available time-series data has been used to train these models. It is thus difficult to verify that the test set is not contaminated by time series related to those in the training set. In contrast, as far as we know, no trajectories generated by classical chaotic systems (e.g., Lorenz equations) have been used to train foundation models. Thus, time series from chaotic systems constitute an independent test set that can be used to quantify the generalization ability of foundation models. Second, chaotic dynamical systems have well-defined attractors that exhibit invariant statistical and geometric properties (fractal dimensions, Lyapunov exponents, power spectra, etc.). This allows us to quantify ML models' ability to capture the long-term behavior of the system even after point forecasts inevitably fail (Pathak et al., 2018; Hess et al., 2023). Such

---

$^*$Correspondence to `gilpin@chaos.utexas.edu`

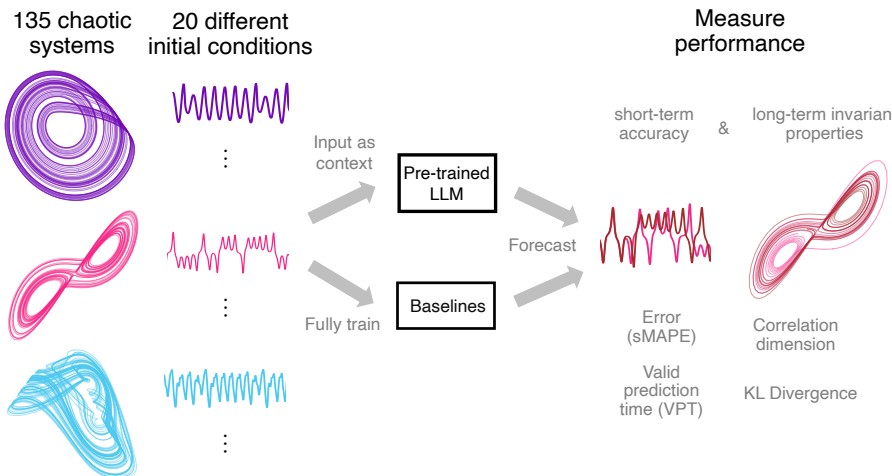

Figure 1: **Chaos as a benchmark for zero-shot forecasting of time series**. We use 135 distinct chaotic systems to generate chaotic trajectories from 20 different initial conditions each. Each trajectory is used to train the baseline deep-learning models (NBEATS, TiDE, etc.) and also provided as context to the pre-trained LLM (we use Chronos, a best-in-class foundation model for time series). Both the trained baseline models and Chronos are then asked to predict the trajectory into the future. We measure the quality of the predictions in terms of both short-term accuracy and long-term attractor reconstruction. Across $10^4$ distinct trajectories and $10^8$ data points, we find that zero-shot forecasts can be competitive in both short-term predictions and in capturing the long-term "climate" of the dynamics.

tests are usually not possible for general time series. Third, the past few years have seen growing activities at the interface of physics and ML (Yu & Wang, 2024; Levine & Tu, 2024; Gilpin, 2024), with the cross-fertilization between ML and dynamical systems yielding advances in both directions (Weinan, 2017; Chen et al., 2018; Pathak et al., 2018; Li et al., 2020; Chen & Tao, 2021; Jordan et al., 2021; Gauthier et al., 2021; Levine & Stuart, 2022; Mikhaeil et al., 2022; Krishnapriyan et al., 2023; Yang et al., 2024). Benchmarking foundation models on chaotic systems introduces the possibility of applying dynamical systems techniques (e.g., Takens embedding theorem (Huke, 2006)) to understand the inner workings of these models and the origin of their generalization abilities.

In this paper, we set out to perform the first systematic evaluation of the zero-shot learning paradigm in the context of forecasting chaotic systems. A schematic summarizing our benchmark pipeline is presented in Fig. 1. We also show another schematic illustrating the difference between classical deep learning models and foundation models when making time series predictions (see Fig. 7 in the appendix).

Our study is also of intrinsic interest to scientific machine learning (SciML) and nonlinear dynamics communities. So far, the data-driven modeling approaches developed in these communities (e.g., reservoir computing (Pathak et al., 2018), PINN (Karniadakis et al., 2021), SINDy (Brunton et al., 2016), Koopman operators (Brunton et al., 2022), neural operators (Azizzadenesheli et al., 2024), etc.) follow a classical train/test dichotomy. That is, to forecast the dynamics of the Lorenz oscillator, a SciML model is first trained on data generated by the Lorenz equations. The model learns chaotic dynamics by extracting the underlying vector field (or flow map) from time-series data during training. At first glance, it seems ludicrous that a model can effectively forecast chaotic dynamical systems without first explicitly learning the flow. A convincing demonstration of the possibility of zero-shot learning in a SciML context could lead to new forecasting tools and generate novel insights into chaotic systems.

From a theoretical standpoint, an emerging direction in SciML is to understand the out-of-distribution generalization ability of different data-driven modeling frameworks (Wang et al., 2020; Kong et al., 2021; 2023; Göring et al., 2024). This parallels a long line of research that investigates the generalization ability of neural networks (Neyshabur et al., 2018; Belkin et al., 2019; Baldassi et al., 2020; Xu et al., 2020; Feng & Tu, 2021; Nakkiran et al., 2021; Power et al., 2022; Liu et al., 2022c). For example, if a model was only trained on trajectories from a limited number of initial conditions, can it effectively extrapolate the learned dynamics to a different part of the phase space

and forecast from a previously unseen initial condition (that is, far from any of the training initial conditions) (Zhang & Cornelius, 2023)? Foundation models that not only generalize to new initial conditions but also to new systems could introduce novel ideas and insights into this endeavor.

Our main contributions are:

1. A large-scale evaluation of the ability of time series foundation models to model physical systems outside of their training domain.

2. Discovery that foundation models produce zero-shot forecasts competitive with models custom-trained to forecast chaotic attractors. Moreover, larger foundation models produce better forecasts.

3. Observation of scaling of a foundation model's zero-shot prediction ability with context lengths far exceeding typical correlation timescales of chaos, indicating in-context learning of chaotic dynamics.

4. Observation that foundation models retain long-term statistical properties of chaotic attractors, even after pointwise predictions fail.

## 2  RELATED WORK

Several works train transformers to perform long-horizon forecasting tasks (Li et al., 2019; Zhou et al., 2021; 2022; Liu et al., 2022b; Wen et al., 2022), obtaining leading results in long-horizon forecasting. However, recent works question their consistency and utility compared to properly-tuned simpler models (Lara-Benítez et al., 2021; Zeng et al., 2023; Das et al., 2023; Tan et al., 2024). Despite these debates, a unique property of large models like transformers is zero-shot generalization, in which they learn to perform a novel task without training the model weights on task-specific data (Brown, 2020). The resulting *in-context learning* strongly differs from prior approaches to forecasting chaotic systems, which focus on training the weights of models based on the past observed history of a system (Pathak et al., 2018; Gauthier et al., 2021; Vlachas et al., 2020). In-context learning has motivated the development of foundation models: large models pre-trained on vast amounts of data, which perform few-shot inference via prompting (Bommasani et al., 2021).

Several recent zero-shot forecasting models are modifications of large language models, which encode time series as tokens (Xue & Salim, 2023; Ansari et al., 2024; Gruver et al., 2024; Miller et al., 2024; Liu et al., 2024b; Ekambaram et al., 2024). Several of these models have been shown to exhibit in-context learning at test time (Lu et al., 2024; Gao et al., 2024; Liang et al., 2024).

Foundation models have recently been introduced for other scientific machine-learning tasks (Miller et al., 2024). These include models for partial differential equations (Yang et al., 2023; Rahman et al., 2024; Subramanian et al., 2024; Herde et al., 2024; Takamoto et al., 2022), numerical integration (Song et al., 2024), fluid flow prediction (Herde et al., 2024), molecular dynamics (Allen et al., 2024), weather forecasting (Nguyen et al., 2023; Bodnar et al., 2024), material discovery (Takeda et al., 2023), astrophysics (Parker et al., 2024), and electrocardiogram (ECG) analysis (McKeen et al., 2024). A recent study used an open-source language model to evaluate zero-shot forecasting performance on stochastic dynamical systems (like Markov chains) as well as the Lorenz attractor (Liu et al., 2024a), finding evidence of a neural scaling law relating context length and prediction accuracy, consistent with prior works (Gruver et al., 2024; Jin et al., 2023). However, to the best of our knowledge, our work is the first large-scale evaluation of the zero-shot learning ability of foundation models on over 100 chaotic systems, both in terms of short-term forecast accuracy and long-term attractor reconstruction performance.

## 3  A MOTIVATING EXAMPLE

Here, we chose Chronos (Ansari et al., 2024) to represent pre-trained models because it has been shown to outperform earlier foundation models for time series, such as TimeGPT and Moirai (Garza & Mergenthaler-Canseco, 2023; Woo et al., 2024). Chronos internally uses a large language model based on the text-to-text T5 transformer model family (Raffel et al., 2020). It introduces a scaling and quantization layer, which converts continuous-valued univariate time series into a set of discrete tokens, with vocabulary size acting as a model hyperparameter. The model was trained on diverse

time series spanning $\sim 10^{11}$ observations drawn from $42$ synthetic and real-world settings, but the training data does not contain any dynamical systems. We evaluate five pre-trained variants of Chronos, denoted by the sizes of the underlying T5 architecture: $8M$, $20M$, $46M$, $200M$, and $710M$ parameters.

Figure 2 shows zero-shot forecasting of the Lorenz oscillator (defined in the appendix), a well-studied chaotic dynamical system, using Chronos-200M. The only data available to Chronos are $512$ data points that serve as the context for the prediction (gray in the Figure). Because Chronos is a univariate forecast model, we separately forecast each coordinate of the attractor and reset the model state between each forecast. Forecasting chaotic systems based on partial observations (e.g., only having access to the $x$ or $y$ coordinate of the Lorenz oscillator) is an extremely difficult task in nonlinear dynamics (Ratas & Pyragas, 2024). Despite this challenge, the prediction closely tracks the ground truth for over 3 Lyapunov times and, even after diverging from it due to chaos, remains in the vicinity of the strange attractor.

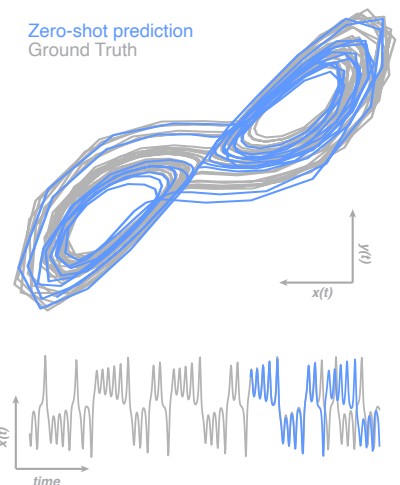

Interestingly, although Chronos predicts $x$ and $y$ separately, it maintains a positive correlation between $x$ and $y$ so they have the same sign most of the time (which is necessary for accurately reconstructing the attractor). This suggests that Chronos internally models $y$ when forecasting $x$ and vice-versa. In nonlinear dynamics, this process is possible due to Takens' theorem, which states that low-dimensional measurements can reveal unmeasured dynamical variables using delay embedding (Huke, 2006).

**Figure 2:** **Zero-shot forecasts of chaotic systems**. We use Chronos to predict the $x(t)$ and $y(t)$ components of the Lorenz oscillator. The zero-shot forecasts match remarkably well with the ground truth for both short-term prediction and long-term attractor reconstruction.

However, the performance of Chronos, while impressive, can also be fragile. Keeping everything unchanged, simply starting the context trajectory from a different initial condition on the attractor can significantly degrade the accuracy of Chronos's prediction (see Figs. 5, 8 and 9). So how good is Chronos at forecasting chaotic systems, truly? More generally, is zero-shot forecasting from foundation models a promising alternative to custom-trained models when it comes to predicting chaotic systems? To answer these questions, we next perform systematic benchmarks that average over a diverse set of chaotic systems and different initial conditions.

## 4 METHODS

**A chaotic systems forecasting benchmark.** The `dysts` dataset represents a standardized benchmark of 135 low-dimensional chaotic systems, described by ordinary differential equations that have been aligned with respect to their dominant timescales and integration steps (Gilpin, 2021; 2023). Each system is annotated with its largest *Lyapunov exponent* $\lambda$, an invariant property associated with every set of differential equations that quantifies the rate at which small errors accumulate. Systems that approach a periodic orbit or an equilibrium exhibit zero or negative Lyapunov exponents because different initial conditions converge to the same state. In contrast, chaotic systems exhibit positive Lyapunov exponents, implying that small changes in the initial conditions or the model parameters lead to trajectories that (at least initially) diverge exponentially. When modeling such systems, a small error will compound over a characteristic timescale, the Lyapunov time, $\tau \equiv \lambda^{-1}$, making highly-chaotic systems (those with small $\tau$) difficult to forecast.

The dynamical attractor of each system in `dysts` is also annotated with mathematical properties such as entropy or fractal dimension. Here, in order to match the typical granularity of the real-world time series used to train Chronos, we re-integrate all systems using an implicit Runge-Kutta integration scheme. We downsample the resulting time series to a uniform coarse granularity of 30 timepoints per Lyapunov time $\tau$. We find that our forecast results depend only weakly on the data granularity (Appendix).

**Baseline experiments.** Our baseline experiment design matches prior works (Gilpin, 2021; 2023; Godahewa et al.; Schötz et al., 2024). For each of the 135 chaotic dynamical systems, 20 trajectories of length 812 are generated, each originating from a random initial condition on the attractor. This produces a set of 2700 ($135 \times 20$) multivariate time series, which have dimensionalities between 3 and 6 depending on the particular dynamical system. All time series are then split into training sets consisting of the first 512 points of each time series, with the last 300 timepoints set aside to determine final test scores. For experiments with varying context lengths, trajectories are extended backwards in time, so that the 300 test points remain the same.

For the baseline models, hyperparameter tuning is performed separately for each of the 135 dynamical systems. For a given dynamical system, each of the 20 training trajectories is divided into a true training set comprising the first 435 timepoints, and a validation set of the last 77 timepoints. For each set of hyperparameters, a model is trained on the true training set and then evaluated on the validation set. The validation scores are averaged over the 20 trajectories, and the hyperparameters from the best-performing model are selected. A model is then initialized with those hyperparameters, and it is trained on the full 512 timepoints. The model is then tasked with autonomously generating a forecast of the next 300 timepoints (around 10 Lyapunov times), which are compared against the ground-truth trajectories to generate overall model scores. The testing dataset is therefore causally disconnected from the training data at all times.

To match the design of Chronos, for multivariate dynamical systems, each baseline model is separately trained and tested along each dimension, and the results are averaged. This channel-independent forecasting task is intrinsically harder than providing full state information, because the models cannot leverage the mutual information between different dimensions (Ratas & Pyragas, 2024). However, recent works on large-scale forecast models actually obtain stronger results by isolating input channels, because the resulting model class is more expressive (Nie et al., 2023). We thus do not expect Chronos's performance to improve if it were instead trained to produce multivariate forecasts (i.e., one in which $x, y, z$ are jointly embedded and tokenized).

The experiments yield 2700 distinct forecasts of 300 timepoints each along $3 - 6$ dimensions depending on the underlying chaotic system, all generated by separately-trained forecast models. Our large-scale experiments thus span $5.5 \times 10^7$ training points, $3.2 \times 10^7$ test points, and $3.2 \times 10^8$ generated forecasts across all models. The experiments require $10^4$ walltime compute hours on an Nvidia A100 GPU.

Our baseline models include NBEATS (Oreshkin et al., 2019), a hierarchical neural network model that has been shown to perform particularly well on dynamical systems forecasting tasks (Gilpin, 2021; 2023). TiDE (Das et al., 2023), a recent model that addresses several known computational limitations of Transformer class models on forecasting time series. A next-generation reservoir computer (NVAR) (Gauthier et al., 2021), which has a strong inductive bias for learning dynamical systems and which has previously been found to perform well on chaotic systems (Gilpin, 2023). We also include a small encoder-decoder Transformer with $0.5M$ trainable parameters, as well as an LSTM (Vaswani et al., 2017; Hochreiter, 1997).

In principle, the baseline models have a wide variety of additional hyperparameters available to tune, such as optimizer settings, reservoir or recurrent layer size, etc. Here, we focus on the lookback window, which is a common hyperparameter across all forecast models. It is also analogous to the context window in Chronos, for which we tune no other hyperparameters in the zero-shot setting.

**Metrics.** Following prior studies (Hyndman & Koehler, 2006; Makridakis et al., 2022; Gilpin, 2021; 2023), we use four metrics to evaluate forecast quality, including *Symmetric Mean Absolute Percentage Error (sMAPE)*.

$$\text{sMAPE}(\mathbf{x}, \hat{\mathbf{x}}) \equiv 2\frac{100}{T} \sum_{t=1}^{T} \frac{|\mathbf{x}_t - \hat{\mathbf{x}}_t|}{|\mathbf{x}_t| + |\hat{\mathbf{x}}_t|},$$

where $\mathbf{x}_1, \mathbf{x}_2, ..., \mathbf{x}_T$ correspond to the true test values of a time series up to a maximum forecast horizon $T$, and $\hat{\mathbf{x}}_1, \hat{\mathbf{x}}_2, ..., \hat{\mathbf{x}}_T$ are the predictions of a forecast model at those same timepoints.

*Valid Prediction Time (VPT).* The first forecast horizon at which the sMAPE exceeds a fixed threshold $\epsilon$ (Vlachas et al., 2020).

$$\text{VPT} \equiv \text{argmax}_{t_f} \{t_f | \text{sMAPE}(\mathbf{x}_t, \hat{\mathbf{x}}_t) < \epsilon, \ \forall t < t_f\}. \tag{1}$$

We set $\epsilon = 30$, as in prior studies (Vlachas et al., 2020; Gilpin, 2023).

*Correlation Dimension ($d_{frac}$).* For chaotic dynamical systems, the long-term distribution of observed data points approximates a fractal object known as the strange attractor. Fractals have space-filling properties that are intermediate between integer dimensionalities, and every strange attractor has a unique and invariant fractal dimension. The correlation dimension non-parametrically estimates the fractal dimension from a time series, by calculating the scaling of the number of other attractor points that fall within a given radius of each point (Grassberger & Procaccia, 1983). We compute the correlation dimension using all data points from a model's forecasts and report the root mean square error between the inferred correlation dimension and the ground truth.

*Kullback–Leibler Divergence between attractors ($D_{stsp}$).* We compute the KL Divergence between the original and reconstructed attractors, following previous works (Hess et al., 2023; Göring et al., 2024). To perform the computation, we center a Gaussian distribution at each point from the true and reconstructed trajectories. We then use a sampling-based approach to estimate the KL Divergence between these Gaussian mixtures (Hershey & Olsen, 2007). This metric measures whether two attractors have matching distributions, and it largely agrees with the correlation dimension. We thus report the KL Divergence results in the Appendix.

## 5 RESULTS

### 5.1 ZERO-SHOT MODELS ARE COMPETITIVE WITH FULLY-TRAINED MODELS IN SHORT-TERM ACCURACY.

To evaluate the effectiveness of zero-shot forecasting for chaotic systems, we evaluate the performance of Chronos and the baseline models on the `dysts` benchmark (Fig. 3). Across the 135 systems, the median VPT of the three largest zero-shot Chronos models is statistically indistinguishable, while the smaller models exhibit significantly smaller VPT ($p < 10^{-3}$, non-parametric Friedman test, $N = 135$). Scaling of performance with model size indicates that the larger models exhibit better generalization properties, because the chaotic systems dataset strongly differs from their training data. This finding supports the premise of the foundation model paradigm for chaotic systems, because it shows that the sheer scale of a domain-agnostic model, when matched with sufficient training, improves forecasts. Compared to the fully-trained baseline models, the three largest zero-shot forecast models outperform all except for NBEATS (Friedman, $p < 10^{-3}$, $N = 135$). While recurrent neural networks and next generation reservoir computers have previously shown promising forecast results for dynamical systems (Vlachas et al., 2020; Gilpin, 2021; Gauthier et al., 2021), they underperform zero-shot models in the data-limited setting investigated here. However, when given enough training data, it has been shown that these models can achieve longer prediction horizons (Gauthier et al., 2021; Gilpin, 2023; Pathak et al., 2018). In contrast, the context length of Chronos and other attention-based forecast models is limited, and they are most effective when data is scarce.

We emphasize that the similarity of the error curves in Fig. 3 does not arise from a lack of sensitivity in the forecast metrics. When the baseline models are instead given full state information (multivariate forecasting), the prediction task becomes easier, resulting in lower sMAPE and higher VPT across all systems (see Appendix). These results underscore that partial observability, which characterizes most practical forecasting tasks (Ratas & Pyragas, 2024), is quantifiably harder for current forecasting models. The zero-shot models perform nearly as well as state-of-the-art, fully-trained models in this setting, reaching a VPT as high as 1 Lyapunov time.

Historically a prediction time of 1 Lyapunov timescale has been considered prohibitive even for fully-trained forecast models. This is because both observational and modeling error compound over this timescale (Palmer, 2000; Medio & Lines, 2001). Chronos's ability to consistently forecast up to 1 Lyapunov time, without prior training on dynamical systems, suggests the advantages of its large-scale training on diverse time series. This scale allows it to extract generic predictive features from time series, which also prove to effectively represent nonlinear dynamics. A similar concept occurs in computer vision, in which convolutional neural networks tend to learn generic Gabor-like feature extractors in early convolutional layers (Zeiler & Fergus, 2014). The ability of Chronos to generate meaningful forecasts suggests that these learned nonlinear features, coupled with high dimensionality both in the input feature space (context length) and internal model dynamics, mitigate

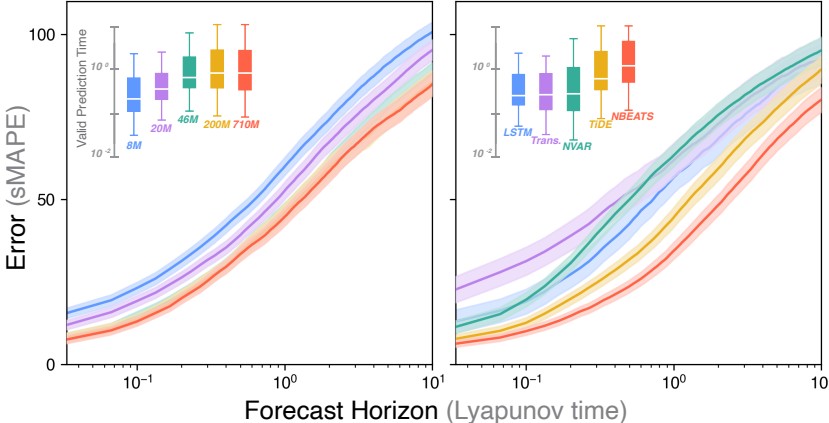

Figure 3: **Zero-shot models of chaotic systems are competitive with custom-trained models**. Zero-shot forecasts from Chronos for five different model sizes (left), compared to other forecast models directly trained on the points given to Chronos as context (right). Inset plots show the valid prediction times (VPT), the first time each forecast exceeds an error limit. All error bars are over 135 chaotic systems, each with 20 distinct initial conditions.

the intrinsic chaoticity of the underlying systems. In dynamical systems theory, recent works on Koopman operators show that appropriately-selected nonlinear transformations make chaotic systems appear more linear (and thus predictable) in higher-dimensional spaces (Mezić, 2013; Brunton et al., 2022). As Chronos contains tens of millions of internal weights, it has an intrinsic advantage due to its scale, which counteracts its low inductive bias when compared to forecasting models specialized for dynamical systems, such as next-generation reservoir computers.

## 5.2 LARGE ZERO-SHOT MODELS EXCEL AT LONG-TERM ATTRACTOR RECONSTRUCTION.

Next, we quantify Chronos and the baseline models' ability to capture the long-term behavior of chaotic systems after point forecasts inevitably fail. This corresponds to a global measure of forecast quality: how well does a model capture the shape of the strange attractor and reproduce the statistics of major dynamic events, even if not necessarily their particular timing? In forecasting, this problem is known as predicting the climate, rather than the weather (Patel et al., 2021; Bramburger & Fantuzzi, 2024).

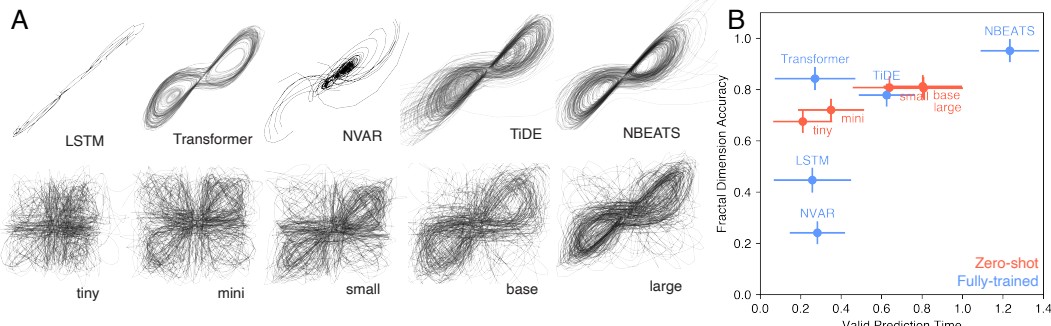

Figure 4: **Zero-shot forecast models effectively capture attractor geometry**. (A) Example forecasts produced by the zero-shot and trained models, for 20 initial conditions from the Lorenz chaotic attractor. (B) The correlation between the fractal dimension of the predicted attractor and the true attractor (Spearman's rank-order coefficient, $N = 2420$ points, $p < 10^{-3}$ for all cases), versus the VPT of the corresponding model. The red markers represent variants of Chronos with different model sizes: tiny ($8M$ parameters), mini ($20M$), small ($46M$), base ($200M$), and large ($710M$). The blue markers represent the baseline models. Models closer to the top capture the attractor geometry better and models closer to the right make accurate point forecasts for longer. Error bars are standard errors over 135 dynamical systems, each with 20 different initial conditions.

Figure 4 shows the correlation dimension accuracy (long-term attractor reconstruction quality) against the VPT (short-term forecast quality) for different models. NBEATS performs the best in both metrics, likely because very high pointwise accuracy necessarily leads to high global attractor quality. Generally, this trend holds across both zero-shot models and baseline models. However, within each model class a few surprises emerge: the fully-trained small Transformer, which produced relatively weak forecasts, captures the attractor shape as accurately as the zero-shot models. This observation suggests that attention-based models, which process their entire context simultaneously, have an innate advantage in capturing the long-term structure of attractors—mirroring similar results for language models (Brown, 2020). Consistent with this interpretation, we observe weak attractor reconstruction accuracy from the LSTM and NVAR models, which both operate sequentially and downweight earlier parts of their context. To ensure that these results are not a consequence of our choice of metric, we also evaluated attractor reconstruction quality using the KL Divergence between the true and forecasted attractors, and we found the same trends (see Appendix).

### 5.3 Zero-shot forecasts parrot motifs from their context.

We next identify a simple mechanism for zero-shot forecasting. Because Chronos is a generative time series model that learns conditional dependencies among points in its context, we directly quantify the similarity between the timepoints immediately preceding a forecast and previous intervals seen in the context. We use the highest-correlating subsequence of duration greater than 30 timepoints (1 Lyapunov time in our units) as a measure of *context overlap*. We find that the zero-shot model's forecasts strongly correlate with this metric over all dynamical systems, and that this dependence is more pronounced than in the best-performing fully-trained model (Fig. 5). This suggests that much of Chronos's performance arises from its ability to parrot context sequences, underscoring our earlier observation that Chronos primarily models conditional dependencies among timepoints.

In Appendix E, we further probe this effect by showing that zero-shot performance continuously degrades as nonstationarity is introduced into the time series. Nonstationarity represents distribution shift for time series, and it disrupts the effectiveness of context-parroting as a forecast strategy because the dynamical attractor continuously and irreversibly changes. In Appendix C, we also identify a weak correlation between forecast accuracy and the first forecast point's natural measure density (the local density of points on a dynamical system's attractor), underscoring how rarer dynamics challenge zero-shot predictions.

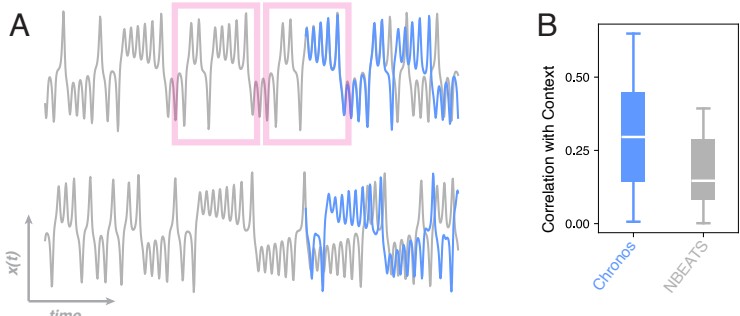

Figure 5: **Context parroting as a mechanism for zero-shot forecasting.** (A) Better zero-shot forecasts often have initial stages that overlap with the context. The context overlap quantifies the similarity between the last 30 points of the context and the prior points. (B) Comparison of context overlap of the zero-shot forecasts (Chronos-base) with the best performing fully-trained model (NBEATS). The zero-shot model correlates with context significantly more than the trained models across the chaotic systems dataset (matched t-test, $N = 135$, $p < 10^{-3}$).

### 5.4 Chronos performs effective in-context learning even with shuffled context.

Chronos's forecasting performance stems from its ability to perform in-context learning, in which early context points on an attractor act analogously to prompts in language models (Brown, 2020; Li et al., 2023). This mechanism underlies our earlier observation that the model's generalization ability improves with its size. While early points in a long context are decorrelated with the predictions,

they are drawn from the same underlying distribution, and we thus hypothesize that longer contexts provide information about the distribution of attractor states, as occurs in language models (Xie et al., 2022). We test this hypothesis by randomly shuffling all length-$k$ sequences of successive timepoints in the model's context, and then repeating our zero-shot experiments as $k$ increases (Fig. 6A). For example, if the context is $\mathbf{x}_1, \mathbf{x}_2, \mathbf{x}_3, \mathbf{x}_4$, then a 1-gram shuffle would be $\mathbf{x}_1, \mathbf{x}_4, \mathbf{x}_2, \mathbf{x}_3$ while a 2-gram shuffle would be $\mathbf{x}_3, \mathbf{x}_4, \mathbf{x}_1, \mathbf{x}_2$. We keep the last $k$ context timepoints the same as the original training dataset, but we ensure that the penultimate $k$ sequence differ from the unshuffled context. As a baseline, we also directly perform zero-shot forecasts using only the last $k$ context timepoints.

We find that the model's forecast accuracy increases with the context length, but that, for sufficiently long contexts, random shuffles provide better forecasts than shorter context baselines. Earlier context points thus provide statistical information about the distribution of single timepoint values, as well as conditional probabilities of certain pairs, triplets, et cetera (Xie et al., 2022). The ergodicity of chaotic attractors implies that they have a well-defined stationary distribution of expected states $p(\mathbf{x_t})$, known as the natural measure (Ott, 2002). Long contexts (even when shuffled), beyond the timescale over which the states of a system become decorrelated, facilitate in-context learning of this measure. Consistent with this observation, in Appendix E, we show that non-stationary time series (in which this measure irreversibly deforms) generally lead to worse zero-shot forecasts. This process resembles the warm-up time in reservoir computers, a type of recurrent neural network used for dynamical systems forecasting (Jaeger & Haas, 2004; Pathak et al., 2018). In this setting, extended context allows the reservoir to gradually synchronize with the dynamical system being learned (Lu & Bassett, 2020).

## 5.5 ZERO-SHOT FORECASTING IS COMPUTATIONALLY EFFICIENT COMPARED TO FULLY-TRAINING MODELS.

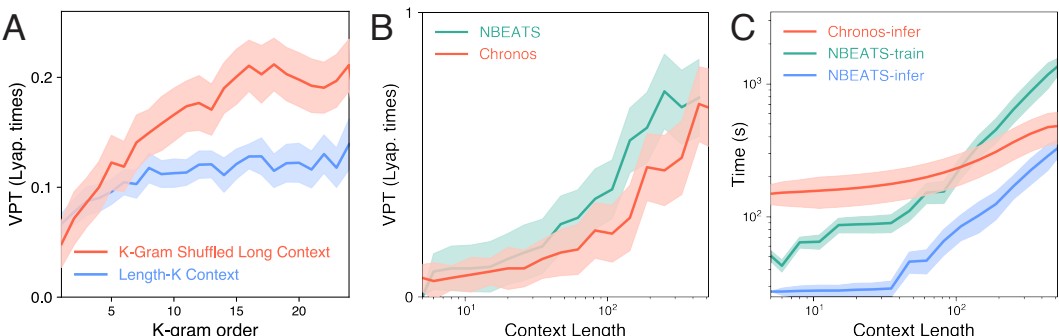

Figure 6: **Scaling laws with context length.** (A) The forecast accuracy (VPT) of Chronos-base when given a context of length $k$ versus when given a full context (length $512$) but with all $k$-grams shuffled. (B) Comparing Chronos-base zero-shot forecasts with NBEATS fully trained on the same context. Both models are trained in a channel-independent manner (C) The single-node walltime for zero-shot forecasts (Chronos-base), compared to the training and inference costs of NBEATS (including hyperparameter tuning). All curves show medians and standard errors over 20 different initial conditions from each of 135 dynamical systems.

We next evaluate how variation in context length affects the performance of Chronos. We vary the context length of the base Chronos model between $5$ and its maximum value of $512$ and repeat our zero-shot forecast experiments. We also select the best-performing traditional model, NBEATS, and fully train it (including cross-validation to set the lookback window) over the same points given to Chronos as context. We find that the VPT of Chronos increases monotonically with context length, even as the context reaches over 17 Lyapunov times (Fig. 6B). This timescale extends well-beyond the $\sim 1$ Lyapunov timescale over which chaotic systems typically become decorrelated ($\langle \mathbf{x}(t)\mathbf{x}(t + \tau)\rangle_t = 0$) (Shaw, 1981). This regime also exceeds the typical range of Takens' embedding theorem, because time series are usually lifted using delay embeddings over timescales $< \tau$. Chronos's performance therefore arises from more than just featurization and extrapolation from recent points in the context.

We next consider the practical question of whether the foundation model paradigm—pretraining in domain-agnostic settings, and then specializing to a task—confers computational advantages over

directly training a smaller model from scratch. We measure the walltime of training and inference on a single A100 GPU node. [1] We find that the computational cost of Chronos can be favorable at long context lengths when compared to NBEATS (Fig. 6C).

The inference time of Chronos is bounded by the quadratic scaling of attention layers with the context length. This limitation motivates newer architectures like Mamba (for language) and TiDE (for time series), which exhibit linear scaling. However, despite the relatively slow inference at small context windows, we find that Chronos can be very efficient when working with long context, making it a viable choice for many practical applications. In terms of the prediction horizon, Chronos exhibits the same linear scaling of cost as auto-regressive models (RC, LSTM, NVAR, etc.).

## 6 CONCLUSION AND FUTURE DIRECTIONS

We have performed the first large-scale evaluation of zero-shot forecasting models on the classical problem of predicting chaos. Our most striking finding is that a large pre-trained model can successfully forecast chaotic systems for up to one Lyapunov time, beyond the expected degree of predictability, even though it was not directly trained on chaotic systems. The resource requirements of inference-only zero-shot forecasting are negligible compared to fully training deep-learning models such as NBEATS, particularly when long context lengths and lookback windows are used. Moreover, zero-shot models perform well without hyperparameter tuning. All in all, the success of Chronos indicates that many aspects of chaotic dynamics can be captured by generic high-dimensional transformations, suggesting that the internal representations used by Chronos to learn dynamical systems may provide insight into other time series tasks, such as system identification or bifurcation detection. It also supports the hypothesis that there is a common "language" for time series—universal features and structures shared by time series across different domains that make transfer learning possible.

On a conceptual level, unpredictability in chaotic systems arises from the rapid growth of the gap between the true trajectory and its approximation by a forecast model—motivating the intuition that Lyapunov time bounds predictability. The long lookback of models like Chronos allows them to leverage information from multiple past timepoints, and thus stabilize accumulation of error relative to forecast models that only consider the most recent timepoints (Viswanath, 2001). In this sense, long-context forecasting resembles multistep integration (Zhang & Cornelius, 2023; 2024). Recent work on dynamic mode decomposition and Koopman operator inference take this idea even further, by showing that time delays can lift dynamical systems into spaces where the dynamics are nearly linear (Brunton et al., 2017). We therefore broadly interpret the zero-shot capabilities of Chronos, which improve with model size, as illustrating the intrinsic inductive bias that comes from lifting nonlinear time series to very high dimensions. However, this does not fully explain our observation that long context windows, spanning multiple Lyapunov times, improve zero-shot forecasts. Instead, we attribute this phenomenon to the recent discovery of in-context learning in pre-trained forecast models, which is only recently starting to be explored in SciML (Yang et al., 2023; Subramanian et al., 2024).

Our study therefore affirms the suitability of the foundation model paradigm for SciML tasks. An important future direction for our investigation is task-specific tuning, in which the weights of large pre-trained models like Chronos are fine-tuned on a small number of example chaotic time series. This differs from the zero-shot in-context learning that we discuss above, and recent foundation models for partial differential equations have found that in-weights tuning can improve generalization (Subramanian et al., 2024). In initial experiments, we found that at least two orders of magnitude more data were required to stably update the weights and validation scores of Chronos. However, this came at the expense of worse performance on the original Chronos training dataset, implying that our dynamical systems dataset differs from a typical time series corpus. This underscores the need for large-scale retraining or low-rank adaptation to further tune Chronos to our task. This mirrors results for large language models, where in-context learning has been shown to be preferable when few examples of the target task are available (Liu et al., 2022a).

---

[1]Walltime imperfectly measures computational costs, as different models are specialized for different hardware (e.g. paralleization or GPU acceleration). Nonetheless, walltime within a given model class provides a proxy for a model's practical performance.

## 7 ACKNOWLEDGMENTS

YZ acknowledges support from the Omidyar Fellowship and NSF DMS 2436231. WG was supported by NSF DMS 2436233 and NSF CMMI 2440490. This project has been made possible in part by Grant No. DAF2023-329596 from the Chan Zuckerberg Initiative DAF, an advised fund of Silicon Valley Community Foundation. Computational resources for this study were provided by the Texas Advanced Computing Center (TACC) at The University of Texas at Austin.

## 8 REPRODUCIBILITY STATEMENT

All zero-shot benchmark forecast results and scripts are available online at https://github.com/williamgilpin/dysts_data. The dynamical systems benchmark dataset is available online at https://github.com/williamgilpin/dysts

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

# A    ADDITIONAL SCHEMATICS

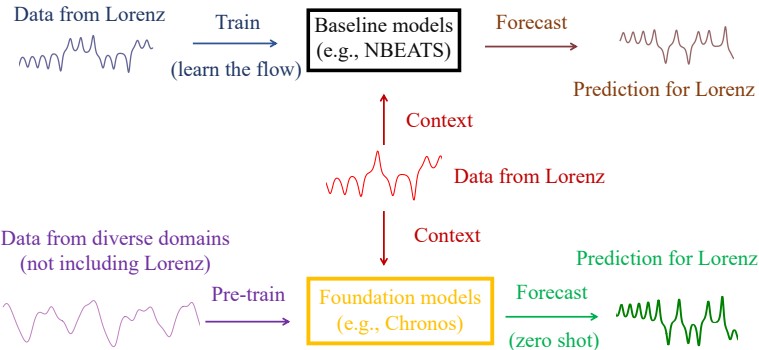

Figure 7: **Difference between baseline models and foundation models in forecasting chaotic systems**. Classical deep-learning models (i.e., baseline models) forecast a chaotic system by learning the underlying vector field or flow map. To achieve this, the model adjusts its weights based on data from the same chaotic system. In contrast, foundation models (e.g., Chronos) do not train directly on the system they want to predict. Instead, they aim to "learn the language of time series" (Ansari et al., 2024) by training on vast amounts of time series data from diverse domains. After that, foundation models can make zero-shot forecasts on any (previously unseen) chaotic system based on a short context trajectory.

# B    ADDITIONAL METHODS

## B.1    LORENZ EQUATIONS.

Lorenz oscillator is one of the most studied chaotic systems and is described by the following equations:

$$\dot{x} = \sigma(y - x),$$
$$\dot{y} = x(\rho - z) - y,$$
$$\dot{z} = xy - \beta z,$$

where the default parameter values are $\sigma = 10$, $\rho = 28$, and $\beta = 8/3$.

## B.2    POINTWISE ERROR METRICS

We quantify point-wise accuracy of forecasts using the *Symmetric Mean Absolute Percentage Error (sMAPE)*.

$$\text{sMAPE}(\mathbf{x}, \hat{\mathbf{x}}) \equiv 2\frac{100}{T} \sum_{t=1}^{T} \frac{|\mathbf{x}_t - \hat{\mathbf{x}}_t|}{|\mathbf{x}_t| + |\hat{\mathbf{x}}_t|},$$

where $\mathbf{x}_1, \mathbf{x}_2, ..., \mathbf{x}_T$ correspond to the true test values of a time series up to a maximum forecast horizon $T$, and $\hat{\mathbf{x}}_1, \hat{\mathbf{x}}_2, ..., \hat{\mathbf{x}}_T$ are the predictions of a forecast model at those same timepoints.

Prior studies have evaluated the suitability of various error metrics in evaluating forecast accuracy (Hyndman & Koehler, 2006; Makridakis et al., 2022), including specifically on dynamical systems prediction (Gilpin, 2021; 2023), and found that sMAPE strongly correlates with other metrics (e.g. RMSE, NRMSE, MASE, Spearman correlation) while exhibiting favorable properties like a bounded range.

## B.3    MEASURING ATTRACTOR SIMILARITY.

We measure attractor similarity using an approach introduced in previous works (Hess et al., 2023; Brenner et al., 2022). The state space divergence between the true and generated attractors is given

by the Kullback-Leibler divergence between the distributions $p(\mathbf{x})$ and $q(\mathbf{x})$,

$$D_{stsp} \equiv D_{\mathrm{KL}}(p(\mathbf{x}) \parallel q(\mathbf{x})) = \int_{\mathbf{x} \in \mathbb{R}^N} p(\mathbf{x}) \log\left(\frac{p(\mathbf{x})}{q(\mathbf{x})}\right) d\mathbf{x}.$$

In high-dimensional spaces, a Gaussian Mixture Model (GMM) is created from the true and generated trajectories in order to approximate these distributions,

$$\hat{p}(\mathbf{x}) = (1/T) \sum_{t=1}^{T} \mathcal{N}(\mathbf{x}; \mathbf{x}_t, \mathbf{\Sigma}_t) \tag{2}$$

and

$$\hat{q}(\mathbf{x}) = (1/T) \sum_{t=1}^{T} \mathcal{N}(\mathbf{x}; \hat{\mathbf{x}}_t, \mathbf{\Sigma}_t).$$

While prior works set the covariance matrix equal to the scaled identity matrix $\mathbf{\Sigma}_t = \sigma_t^2 \mathbf{1}$ with $\sigma_t = 1$ for all $t$, we instead set $\sigma_t = \|\mathbf{x}_t - \mathbf{x}_{t-1}\|$ in order to adjust for uneven spacing among data points. We next perform Monte Carlo sampling and estimate the KL divergence as

$$D_{stsp} \approx \frac{1}{n} \sum_{i=1}^{n} \log\left(\frac{\hat{p}(\mathbf{x}^{(i)})}{\hat{q}(\mathbf{x}^{(i)})}\right),$$

where $\mathbf{x}^{(i)}$ are samples drawn from the true orbit (Hershey & Olsen, 2007).

## C  DEPENDENCE OF FORECAST ACCURACY ON INITIAL CONDITIONS

We investigate the degree to which zero-shot forecasting performance depends on the initial condition. As an illustrative example, in the right panel of Figure 8, we repeat the experiment shown in Figure 2, but for a different initial condition. We use the base Chronos model with a maximum context of 512 points, but we choose a trajectory emanating from a different point on the chaotic attractor. We see that the performance of Chronos is noticeably worse for this trajectory, indicating that the particular choice of initial conditions can influence zero-shot performance.

For both initial conditions, Chronos attempted to perform pattern matching by looking for snippets in the context trajectory that most closely resemble the history immediately preceding the prediction and simply repeating that motif. The difference is that there is a very good repeating pattern in the context trajectory on the left but not on the right, which directly leads to worse prediction from the second initial condition. From the perspective of Takens' embedding theorem, this context-matching strategy is trying to find the closest context data point to the initial condition in the delay embedding space and repeating the context trajectory from that point.

To further quantify variability in forecast accuracy caused by initial conditions, we sample a set of 200 trajectories originating from different points on the attractor, and generated zero-shot forecasts that we evaluated using the VPT (Eq. 1). We define the initial condition for each trajectory as the final point of the context given to the model before a forecast. We observe wide variation in prediction performance with the initial condition (Fig. 9, with a nearly exponential distribution of VPT across initial conditions. Thus while the median VPT of Chronos is relatively high (approaching 1 Lyapunov time for the largest models), occasionally an initial condition will result in a forecast that remains accurate for several Lyapunov times.

In order to identify the origin of these anomalously long forecasts, we calculate the relative denstity of the attractor at each initial condition. Chaotic dynamical systems approach a steady-state distribution of points, the strange attractor, with a continuous density $\mu(\mathbf{x})$ known as the natural measure of the system. We estimate $\hat{\mu}(\mathbf{x})$ using Eq. 2 for each initial condition, and compare the VPT$(\mathbf{x})$ of a forecast originating from each initial condition $\mathbf{x}$ to the estimated measure at that point $\hat{\mu}(\mathbf{x})$. We perform this procedure for 20 distinct initial conditions from each of the 135 chaotic dynamical systems in our dataset (Fig 9B). In the figure, we highlight the initial conditions for the Lorenz attractor in blue. We find a weak but robust correlation between measure and forecast accuracy (Spearman's rank order coefficient, $\rho = 0.26 \pm 0.03$, $p < 10^{-3}$, $N = 2700$. This is consistent with the idea that zero-shot forecast models perform better at forecasting denser, more common regions of the attractor, because those points are more common in the context. Conversely, rarer points (i.e., extremal points closer to out-of-distribution dynamics relative to the context points) lead to worse forecasts.

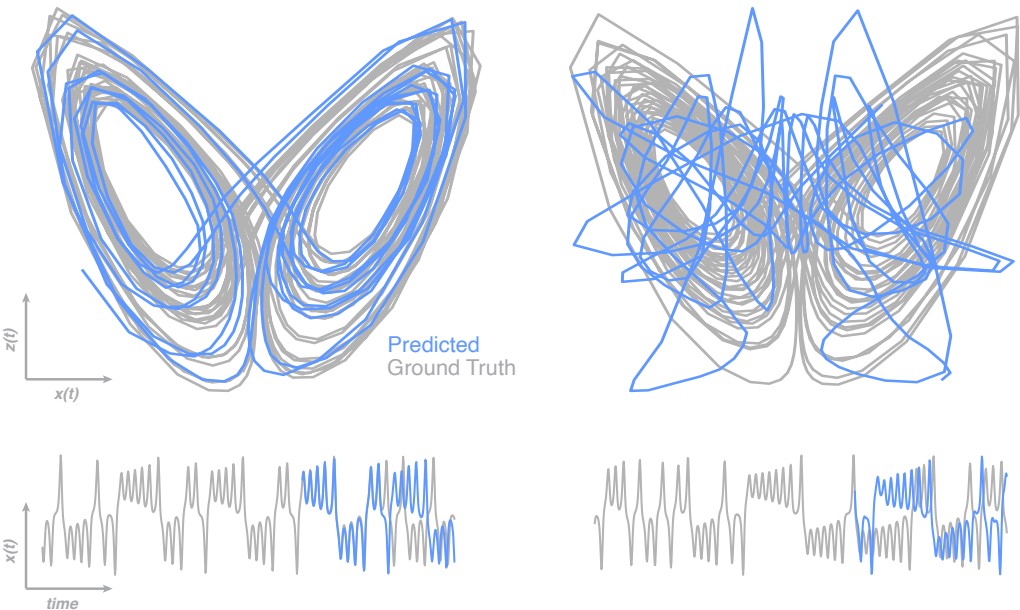

Figure 8: **Zero-shot forecasting performance depends on initial conditions**. Zero-shot forecasts of the Lorenz attractor using Chronos-base for two different initial conditions on the Lorenz attractor. Both forecasts use the same context length of 512 timepoints; their performance difference arises only from their starting point.

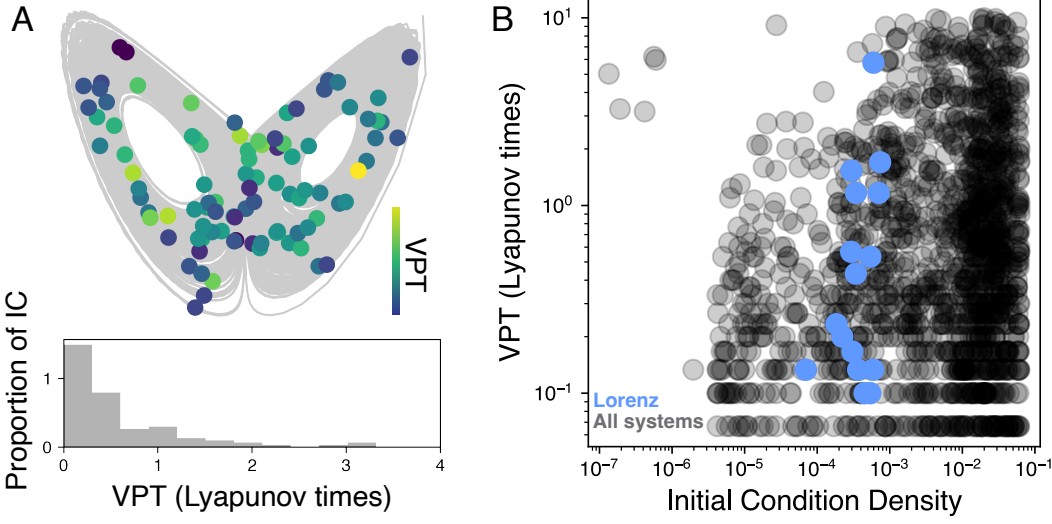

Figure 9: **Quantification of the dependence of zero-shot forecasts on initial conditions**. (A) A set of points on the Lorenz chaotic attractor, colored by the forecast accuracy (VPT) of zero-shot forecasts in which they were the final context point. A histogram of the accuracy values is underlaid. (B) The forecast accuracy (VPT) versus the relative density of the region of the attractor in which the last context point appears. Black circles indicate 20 initial conditions from each of 135 chaotic dynamical systems, and the 20 initial conditions from the Lorenz attractor are highlighted in blue.

# D    APPLICATION TO REAL-WORLD CHAOTIC SYSTEMS

We next compare our results for our large-scale dynamical systems benchmark dataset to real-world multivariate time series from chaotic systems. Unlike simulated differential equations, real measurements exhibit measurement error, stochasticity, and non-stationarity.

Our experimental dataset consists of a 400 fps video of an oscillating double pendulum, as recorded on a high-speed Phantom Miro EX2 camera (Asseman et al., 2018). The video is paired with a time series of centroid coordinates for each pendulum hinge and joint, as extracted by the original authors using object tracking. This time series consists of positions of the pivot attachment to the wall, the hinge connecting the first and second pendula, and the second pendulum's tip. We transform the dataset into new sequences that represent the angles each pendulum forms with the vertical axis, denoted as $(\theta_1, \theta_2)$. We then numerically differentiate these angle measurements to obtain the angular velocities $(\dot{\theta}_1, \dot{\theta}_2)$. In an ideal double pendulum system, the set of four variables $(\dot{\theta}_1, \dot{\theta}_2, \theta_1, \theta_2)$ uniquely parameterizes the Hamiltonian, thereby defining the system's attractor. However, for the experimental data, the time-averaged kinetic energy $T \propto \dot{\theta}_1^2 + \dot{\theta}_2^2$ gradually decreases over the course of the experiment. As a result, the pendulum dataset is non-stationary, with an attractor that gradually changes over time. We downsample this time series by a factor of 3.

We use Chronos (base model) to forecast this dataset for 7 non-overlapping contiguous time intervals spanning the full experiment. Each window is split into a context window of length $512$ and a testing dataset of length $300$, for a total of $8 \times (512 + 300) \approx 6500$ total timepoints. We find that the error exhibits similar scaling as we observe for ergodic dynamical systems in the main text (Fig. 10). This indicates that experimental variation and measurement errors do not preclude the application of zero-shot forecasting to chaotic time series generated by real-world systems. Additionally, the pendulum dataset exhibits non-stationarity due to gradual loss of energy from the system. As a result, this dataset exhibits weak distribution shift between the training (context) and testing (zero-shot forecasting) settings. Because we observe the same general scaling of error as in the $135$ ergodic and stationary systems, we conclude weak distribution shift does not preclude effective zero-shot forecasting. Thus, in this example, Chronos exhibits out-of-domain generalization, because the underlying chaotic attractor (and thus distribution of testing points) changes relative to the context.

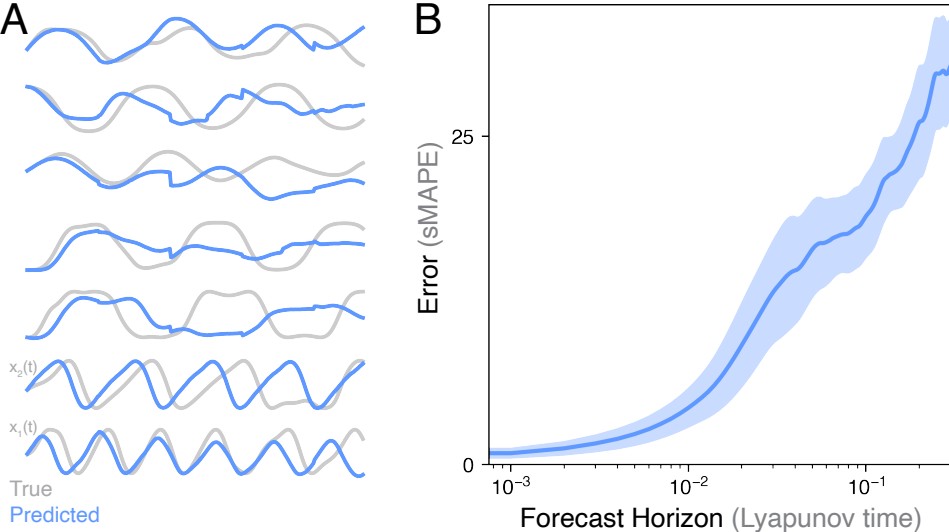

Figure 10: **Zero-shot forecasting of a chaotic pendulum experiment**. (A) Zero-shot forecasts along the first angular coordinate of a double pendulum for the base Chronos model, for 7 different initial conditions. (B) Scaling of forecast error with forecast horizon. Curve corresponds to means and standard errors across 7 initial conditions and 4 coordinates each.

## E  PROBING OUT-OF-DISTRIBUTION DYNAMICS AS TRAJECTORIES LEAVE THE ATTRACTOR

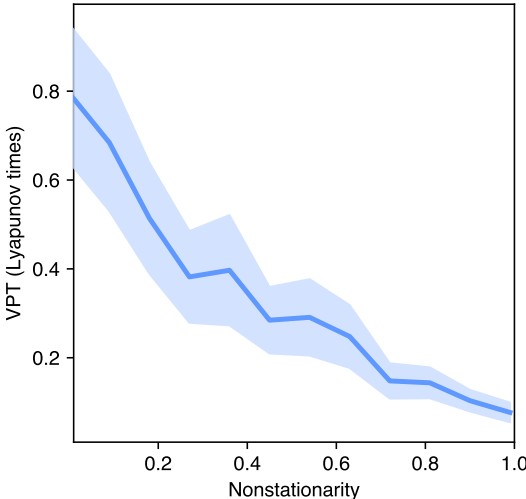

Figure 11: **Zero-shot forecasts degrade with distribution shift**. Forecast accuracy (VPT) of zero-shot forecasts with Chronos-base, as the degree of nonstationarity in the time series varies via Eq. 3. Curve and error bars are median and standard error over 20 initial conditions for each of $N = 135$ chaotic dynamical systems.

We next evaluate the degree to which non-stationarity affects zero-shot forecasting performance. For each trajectory considered in the main text, we apply an exponential modulation along the time dimension. For a time series of length $T$ given by $\mathbf{x}_1, \mathbf{x}_2, ..., \mathbf{x}_t, ..., \mathbf{x}_T$, the exponentially-decaying modulation has the form,

$$\mathbf{x}_t \leftarrow \mathbf{x}_t e^{t \frac{\log f_{\min}}{T-1}} \tag{3}$$

By decreasing $f_{\min}$ from 1 to 0, we increase the degree to which the dynamics appear non-stationary. When $f_{\min} = 1$, then the damping term becomes a constant and the dynamics are unaffected. However, when $f_{\min} \to 0$, the dynamics resemble damped oscillations that monotonically approach a fixed point. We thus consider experiments forecasting time series with $f_{\min} < 1$ a quantitative probe of the degree to which zero-shot forecasts are applicable to real-world systems, in which the chaotic attractor irreversibly deforms due to processes like dissipation. In a machine learning context, this setting corresponds to out-of-distribution or out-of-domain generalization, in which the forecast points describe a different dynamical regime than the context (Göring et al., 2024).

We find that, across all 135 systems, the performance of Chronos degrades as the degree of nonstationarity $1 - f_{\min}$ increases (Fig 11). This observation matches our intuition, based on our observation in the main text that Chronos performs in-context learning of the distribution of points (and pairwise, $k$-wise conditional dependencies among successive timepoints). We also find in the main text that Chronos performs more strongly on trajectories resembling its training data. Nonstationarity undermines all of these mechanisms, leading to the degradation in performance as the forecast regime more strongly differs from the context.

Because context-parroting is a particularly effective strategy for stationary systems like ergodic chaotic attractors, time series models like NBEATS, which can directly identify and model monotonic trends, have an advantage on simple out-of-distribution forecasting tasks like the one we consider here. NBEATS and its variants have successfully been applied to several types of time series with predominant trends, underscoring their advantage in this setting Challu et al. (2023). Based on this observation, we anticipate that several modifications could make foundation models like Chronos more robust to weak nonstationarity: (1) Chronos currently uses an encoder-decoder language model Raffel et al. (2020). Using Chronos's tokenizer in tandem with a modern language model with an explicit positional encoding scheme, like rotary positional embedding, would provide the model with explicit time information that would allow it to capture longer-term trends in

a time series Su et al. (2024). (2) Pretraining with short time series. While Chronos's original training dataset includes many nonstationary processes, shorter time series generally exhibit greater nonstationarity, and so their inclusion represents a simple mechanism to improve model robustness. (3) Biasing generative forecasting towards rarer states. As a generative model, Chronos generates forecasts probabilistically by sampling multiple potential future trajectories. Modifications of this scheme that encourage oversampling of rarer states could help the model better account for irreversible processes, though potentially at the expense of lower performance on ergodic processes.

## F  BASELINE MODELS

### F.1  BASELINE MODEL HYPERPARAMETERS

Our baseline models follow the experiment design and hyperparameter tuning procedure used in prior works on the chaotic systems dataset (Gilpin, 2021; 2023). Those works contain qualitative descriptions of the different models, and the performance results obtained in those works motivate our particular baseline model choices. We also include the Time-series Dense Encoder (TiDE), a newly introduced linear state space model that can achieve nearly-optimal error rates for linear dynamical systems (Das et al., 2023). For many models, we use reference implementations and hyperparameters found in the `Darts` forecasting library (Herzen et al., 2022). For the next-generation reservoir computer (NVAR), we use the default settings used in the original work (Gauthier et al., 2021). However, in order to fairly tune hyperparameters across models, for each model we select one hyperparameter to tune that corresponds to the lookback window, or context, that sets the number of past timepoints that the model simultaneously processes when generating a forecast.

**N-BEATS Model**  (Oreshkin et al., 2019)

- **Key Hyperparameters:**
  - *Input Length*: *Tuned for each system among {0.067, 0.167, 0.333, 0.5, 0.833, 1} Lyapunov times*
  - *Number of Stacks*: 30
  - *Number of Blocks*: 1
  - *Number of Layers*: 4
  - *Layer Widths*: 256
  - *Expansion Coefficient Dimension*: 5
  - *Degree of Trend Polynomial*: 2
  - *Dropout Fraction*: 0.0
  - *Activation Function*: ReLU

**Transformer Model**  (Vaswani et al., 2017)

- **Key Hyperparameters:**
  - *Input Length*: *Tuned for each system among {0.067, 0.167, 0.333, 0.5, 0.833, 1} Lyapunov times*
  - *Number Attention Heads*: 4
  - *Number Encoder Layers*: 3
  - *Number Decoder Layers*: 3
  - *Dimension Feedforward*: 512
  - *Dropout Fraction*: 0.1
  - *Activation Function*: ReLU

**TiDE**  (Das et al., 2023)

- **Key Hyperparameters:**
  - *Input Length*: *Tuned for each system among {0.067, 0.167, 0.333, 0.5, 0.833, 1} Lyapunov times*

- – *Number of Encoder Layers*: 1
- – *Number of Decoder Layers*: 1
- – *Decoder Output Dimension*: 16
- – *Hidden Dimension Size*: 128
- – *Past Temporal Width*: 4
- – *Future Temporal Width*: 4
- – *Past Temporal Hidden*: None
- – *Future Temporal Hidden*: None
- – *Temporal Decoder Hidden*: 32
- – *Dropout Fraction*: 0.1

**NVAR**  (Gauthier et al., 2021)

- **Key Hyperparameters:**
  - – *Number Input Lags*: *Tuned for each system among* {*0.067, 0.167, 0.333, 0.5, 0.833, 1*} *Lyapunov times*
  - – *Maximum Order*: 2
  - – *Regularization*: $10^{-4}$
  - – *Stride*: 1.0

**LSTM**  (Hochreiter, 1997)

- **Key Hyperparameters:**
  - – *Input Length*: *Tuned for each system among* {*0.067, 0.167, 0.333, 0.5, 0.833, 1*} *Lyapunov times*
  - – *Hidden Dimensionality*: 25
  - – *Number of Recurrent Layers*: 2
  - – *Dropout Fraction*: 0.0
  - – *Training Length*: 24

### F.2 Fine-tuning Chronos

As an informative additional baseline, we attempted to fine-tune Chronos-base on the chaotic systems dataset. From the zero-shot experiments, we compiled a collection of $1.3 \times 10^6$ observations, corresponding to trajectories of length 512 timepoints originating from 20 initial conditions for each of 135 chaotic dynamical systems. We fine-tuned Chronos-base using the authors' original training scripts, with all hyperparameters matching those used in the original Chronos training run Ansari et al. (2024). On our zero-shot dataset, we did not observe a strong improvement in Chronos's validation scores on held-out trajectories. Instead, the loss plateaued early during training, and the qualitative appearance of forecasts did not improve over the zero-shot case. When we instead tried only fine-tuning on a single system, the Lorenz attractor, we observed similar results. Moreover, we observe a weak reduction in forecast accuracy on datasets randomly drawn from Chronos's training corpus. Across the 135 chaotic dynamical systems in our dataset, we did not observe a general relationship between fine-tuning performance and invariant properties of the underlying system, such as dimensionality or Lyapunov exponents.

Based on these observations, we conclude that the training behavior of Chronos is decoupled from properties of the underlying datasets in the training regime we reach in our fine-tuning experiments. We thus conjecture that the chaotic systems time series dataset strongly differs from the large time series corpus on which Chronos was originally trained, leading to fine-tuning failing due to strong task shift Kumar et al. (2022). This phenomenon represents a variant of out-of-distribution generalization error, manifesting as slow convergence on new datasets. We therefore expect that fine-tuning Chronos for chaotic systems will require full retraining on a dataset comparable in size to the Chronos training corpus ($10^{10} - -10^{11}$ observations), as well as potential customizations of the tokenizer and language model to better handle dynamical systems datasets. For example, recent works note that multivariate time series often exhibit weak coupling among channels, motivating

the general use of channel-independent training schemes Nie et al. (2023). We also expect that new hyperparameters, particularly training schedule and optimization rates, will need to be selected in order to obtain noticeable improvements. This level of tuning and data scale exceeds that used for the other baseline models, and so we defer further investigation of fine-tuning and few-shot learning to future work. Additionally, in order to avoid fully retraining Chronos for our task, alternative strategies such as low-rank adaptation Hu et al. (2021), and its generalizations for time series forecasting Gupta et al. (2024), may be applied in future work.

# G    ADDITIONAL EXPERIMENTS AND ANALYSES

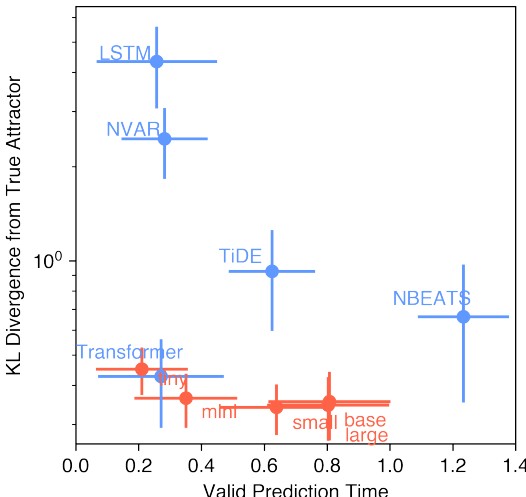

Figure 12: **Zero-shot forecast models capture attractor geometry well, as measured by the KL Divergence**. The state space divergence $D_{stsp}$ between the predicted attractor and the true attractor, versus the VPT of the corresponding model. The red markers represent variants of Chronos with different model sizes: tiny ($8M$ parameters), mini ($20M$ parameters), small ($46M$ parameters), base ($200M$ parameters), and large ($710M$ parameters). The blue markers represent the baseline models. Models closer to the bottom capture the attractor geometry better, and models closer to the right make accurate point forecasts for longer.

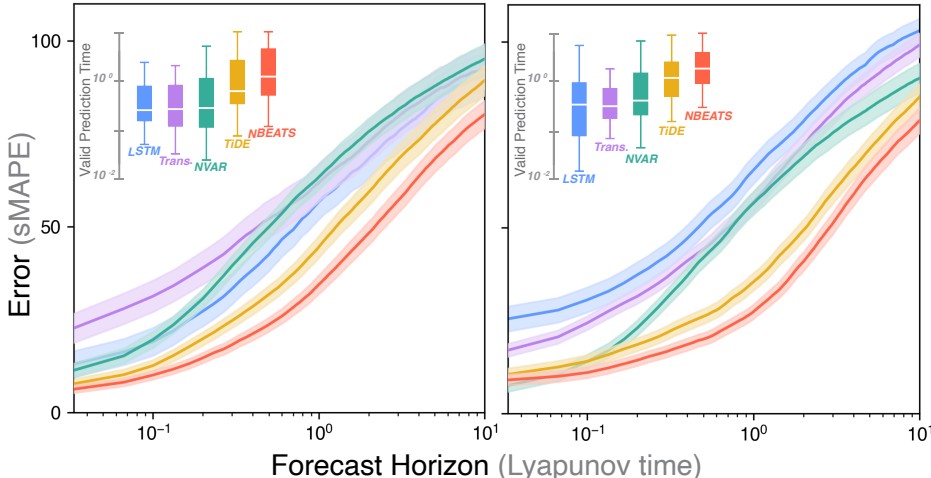

Figure 13: **Comparison of univariate versus multivariate baseline forecasts of chaotic systems**. Because Chronos is a univariate forecast model that predicts each time series channel independently, the baseline experiments we present in the main text (left panel here) involve channel-independent training, in which each baseline model is separately trained and tested on each dimension of the input time series. We repeat these experiments in a multivariate setting, by retraining the baseline models simultaneously on all dimensions (right panel). All error bars are over 20 distinct initial conditions for each of the 135 chaotic systems.

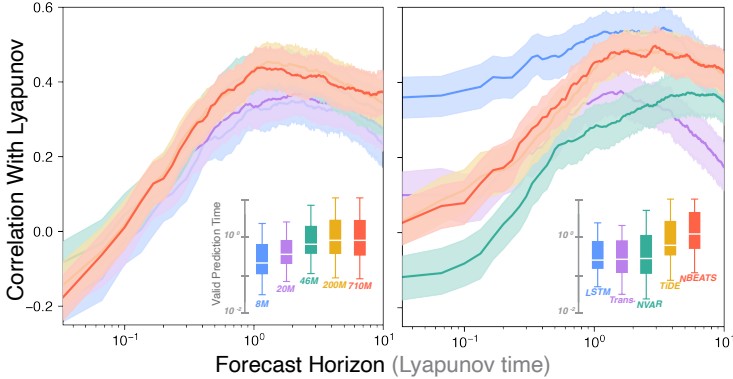

Figure 14: **Correlation between forecasts and invariant properties**. The correlation between the Lyapunov exponent of each of the 135 chaotic systems, and the sMAPE error of a forecast model, as a function of the prediction horizon.

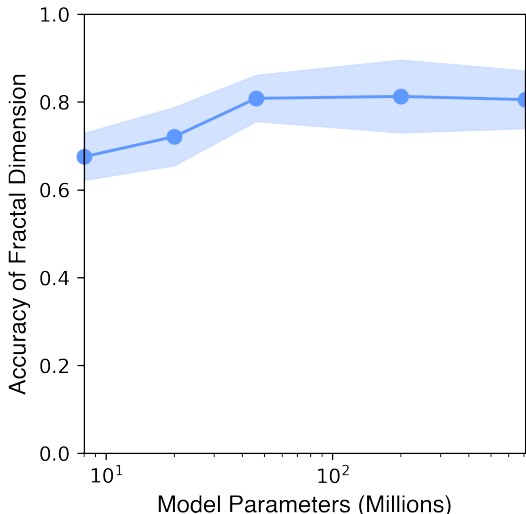

Figure 15: **Zero-shot attractor reconstruction accuracy scales with model size**. The Spearman correlation between the fractal dimension of Chronos's predictions, and the true fractal dimension of the underlying system, compared to the number of trainable parameters in the Chronos model.

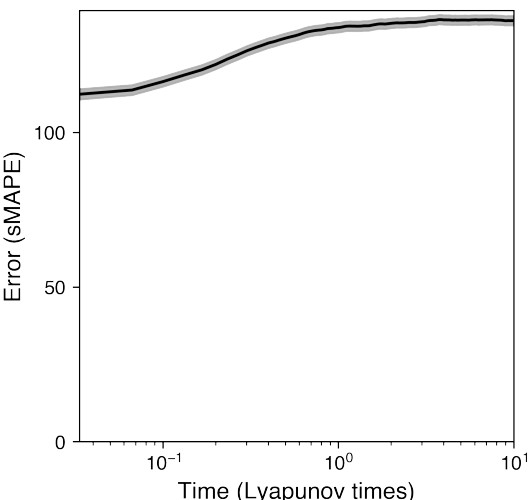

Figure 16: **Naive forecasts underperform all models evaluated.** The growth of the sMAPE error for a naive constant forecast, in which the most recent training point is carried forward as the prediction for all future values. The shaded region corresponds to standard error across 135 dynamical systems, with 20 initial conditions each..

