# OpenReview forum: "Zero-shot forecasting of chaotic systems"
_ICLR.cc/2025/Conference — ICLR 2025 Poster_

### Official Review · Reviewer_MDoC · 2024-10-29

**Soundness:** 3
**Presentation:** 3
**Contribution:** 2
**Rating:** 8
**Confidence:** 3

**Summary:**

The present paper proposes to use time-series generated from chaotic systems as a new benchmark in time-series forecasting.
It makes the case for time-series foundation models (Chronos) which were pretrained on diverse synthetic and real-world data which it finds to yield better results than customly trained algorithms. This is surprising given that Chronos is a univariate prediction model and therefore sees the x, y, z components of the chaotic systems independently.

**Strengths:**

- The core contribution is a benchmark of foundation models on the evolution of different chaotic systems. I think this is a good contribution which goes into a similar direction as Chronos and ForecastPFN which use synthetic data (Chronos not exclusively) to pretrain their models and still obtain good performance. I agree with a reference in the conclusion stating that there appears to be common language shared by time-series as it's surprising that the existing synthetic data generators would result in such well performing models.
- I see potential for this benchmark to be used itself for more expressive synthetic data in time-series foundation models.
- I think the finding that chaotic systems can be learned better by foundation models from context rather than by a specific model is valuable and interesting.

**Weaknesses:**

- It would be good to give more details on the LSTM and transformer models shown in the paper in order to allow for a fair comparison. I couldn't find details on the number of layers used e.g. for LSTM after hyperparameter optimization like the number of layers or the state size of the LSTM.
- Could you clarify your rational on choosing the models you evaluated in the paper? Given the recent interest in linear RNNs like Mamba 1 / Mamba 2 / Gated Linear Attention, could you provide some results in this model class? Maybe Mamba 2, as it's fast to train.
- I think the paper in its current state is quite verbose and could benefit from a clearer structure. For example sections 5.1 and 5.2 could have individual paragraphs with titles indicating the finding which the authors would like to emphasize.

**Questions:**

-

---

> ### Author Response · Authors · 2024-11-21
> **Detailed Reply**
>
> Thank you very much for your positive review and suggestions! We very much agree that it is important to provide experimental details for reproducibility and to convey the results clearly. We have added the requested model details and improved the writing. We hope that you will find these changes responsive; please let us know if you have any additional questions or comments. *Please see the joint response above for a summary of all new experiments and improvements to the paper.*
>
> **[MODEL DETAILS]**
>
> *See new Appendix F*
>
> We have added a new appendix (Appendix F) that specifies all models, their origin and original reference, their default hyperparameters, and the particular hyperparameters that we tuned.
>
> **[MODEL CHOICES]**
>
> Regarding newer models, we tried to stick to time series models that the SciML community typically uses for chaotic systems. NBEATS and reservoir computers are both used frequently in this space. TiDE, which is included among our benchmarks, is a linear recurrent model along the lines of Mamba, but which has an inductive bias for dynamical systems---its authors actually showed that it approaches the theoretical error bound for linear (and thus non-chaotic systems).
>
> Since Mamba 2 is a language model, we decided that adding a Chronos-style tokenizer and retraining it end-to-end was beyond the scope of what we can accomplish to a high standard during the review period. Note that we *separately tune hyperparameters for each of the 135 chaotic systems*, and so adding baseline models requires significant computational overhead.
>
> If the referee feels particularly strongly about this, I'm open to adding a SOTA time series model like Patch-TST in the camera ready phase, but getting this working carefully requires a level of engineering and compute that I feel is impractical during the rebuttal period.
>
> **[PRESENTATION AND CLARITY]**
>
> We have further polished and streamlined the presentation of the paper. We have now broken the "Results" section into five subsections with titles that explicitly summarize the key points we want to convey in each subsection.

---

> > ### Comment · Reviewer_MDoC · 2024-11-23
> >
> > Thank you for improving the clarity of the paper and the comprehensive rebuttal.
> > Given the number of linear RNN based time-series models based on Mamba I would recommend to add Mamba as a baseline because it could make your paper more interesting to this community. I will raise my score slightly, as I think that the paper presents interesting results for the community.

---

> > > ### Author Response · Authors · 2024-11-24
> > > **Thank you**
> > >
> > > Thank you very much for considering our revision and for raising your score. We very much appreciate it. I agree that a baseline like Mamba 2 would be interesting, particularly because Mamba has recently been shown exhibit aspects of in-context learning, which we think underlies Chronos's performance on this task. I'll see if I can get similar behavior to emerge here by using Chronos's tokenizer with Mamba. Thank you very much for reviewing our paper.

---

### Official Review · Reviewer_yZ6s · 2024-10-30

**Soundness:** 2
**Presentation:** 3
**Contribution:** 2
**Rating:** 6
**Confidence:** 4

**Summary:**

The paper presents a large-scale evaluation of the zero-shot learning capabilities of foundation models for chaotic systems. The authors investigate both short-term and long-term forecasting scenarios. They evaluate the foundation models against over 100 chaotic systems and observe several interesting findings. The key point is that foundation models not only generalize to new initial conditions but also to entirely new systems. This suggests the models have the capacity to capture the broader dynamical structures of chaotic systems.

**Strengths:**

The paper presents strong findings on the power of foundation models for zero-shot learning in chaotic systems. Firstly, it demonstrates that foundation models can perform as well as fully trained models across multiple scenarios, especially in long-term forecasting.

Another strength of the paper is its study and evaluation of the suitability of foundation models in the field of scientific machine learning. This proof of concept can definitely lead to new developments in the study of chaotic systems.

**Weaknesses:**

Further experiments are needed to assess sensitivity to initial conditions and real-world chaotic systems, as these factors currently weaken the proposed evaluation and findings.

While the application and findings of the paper are important, I believe the work in its current state lacks novelty due to limited experimental section.

**Questions:**

The authors aim to evaluate the proposed approach on real-world data? Interesting characteristics can be observed.

How does foundations models forecast accuracy vary with different initial conditions within chaotic systems? This sensibility limits its generalization?

I suggest the authors to state limitations of the zero-shot long-term prediction problem. Another possible limitation is how the foundation model zero-shot approach will behave when there is a drastic different dynamical behaviour between training and testing.

---

> ### Author Response · Authors · 2024-11-14
> **Clarifying questions at start of discussion period**
>
> **[Initial clarifying questions at start of discussion period]**
>
> Thank you very much for your review and comments. We were a bit surprised by the low score given the content of the review, which we feel is generally positive.
> **We will address your other comments with additional experiments in our full response later, but can you clarify at this stage why you feel the paper lacks novelty?**
>
> To our knowledge, our paper is the first to apply zero-shot learning to dynamical systems, and in particular to show that zero-shot forecasts can compete with specialized SciML models. As far as we know, there are very few prior works that systematically apply the zero-shot paradigm to SciML tasks, which can potentially open new avenues for solving many important questions in the sciences.

---

> > ### Comment · Reviewer_yZ6s · 2024-11-15
> >
> > Dear authors I agree there is value on the contribution as a initial work on zero-shot learning to SciML tasks. As I stated in the review I think the experimental part is a bit limited but I am willing to raise my score. With the new results I am willing to further raise it.

---

> ### Author Response · Authors · 2024-11-21
> **Detailed Reply**
>
> Thank you very much for your review and for raising your score! In our revision, we have sought to address all of your points with new experiments. Please let us know if you have any additional questions or comments.
>
>
> **[REAL-WORLD DATA]**
>
> *See revised text Fig. 10 and new Appendix D.*
>
> We have now explored Chronos' ability to forecast real-world chaotic systems with an experimental dataset consisting of a 400 fps video of an oscillating double pendulum. The double pendulum experimental data exhibit noise and measurement errors, and also exhibit distribution shift (i.e., non-stationarity) due to the gradual loss of energy from friction. We found that the zero-shot forecasting error exhibits similar scaling as the error for the ergodic dynamical systems in the main text (Appendix D and Figure 10). This suggests that foundation models such as Chronos can make effective forecasts for real-world chaotic systems, even in the presence of noise and weak distribution shift.
>
> **[SENSITIVITY TO INITIAL CONDITIONS]**
>
> *See revised text Fig. 5, Fig. 9, Fig. 11 and new Appendices C and E.*
>
> Our new experiments answer the question: how does initial condition variability impact zero-shot forecasting accuracy? In general, we confirmed that there is significant variability of forecast accuracy depending on the initial condition (spanning over two orders of magnitude of the VPT, see new Fig. 9). Moreover, we found a weak but robust correlation between the natural measure (density of training points) on the attractor and forecast accuracy (See new Appendix C and Fig. 9). This is consistent with the idea that zero-shot forecast models perform better at forecasting denser, more common regions of the attractor, because those points appear more often in the context. Conversely, rarer points (i.e., extremal points closer to out-of-distribution dynamics relative to the context points) lead to worse forecasts.
>
> Most interestingly, we found that there is a much stronger correlation between context overlap and forecast accuracy (Fig. 5). If you look at the first trajectory in Figure 5, you will see that Chronos seems to be copying and pasting part of the context trajectory in the forecast. This strategy (looking for snippets in the context trajectory that most closely resemble the history immediately preceding the prediction and simply repeating that pattern) is extremely simple but can actually be very effective. You can see Chronos trying to do the same thing for the second trajectory, only that this time it cannot find a very good repeating pattern in the context trajectory, so the prediction is worse. From the perspective of Takens’ embedding theorem, this strategy is trying to find the closest context data point to the initial condition in the delay embedding space and repeating the context trajectory from that point. We hypothesize that this is one of the strategies used by Chronos for zero-shot forecasting, which led us to compute the **Context Overlap**, a measure of minimum distance between the context points and the initial condition in the delay embedding space. Across $20$ initial conditions from $135$ systems (Fig. 5), we found that context overlap strongly predicts forecast accuracy, which explains the observed variability of forecast performance on initial conditions and lend credence to the idea of context parroting is a mechanism for zero-shot forecasting.
>
> **[OUT-OF-DISTRIBUTION GENERALIZATION]**
>
> *See revised text Appendix E and Figure 11.*
>
> Our new experiments probe the performance of zero-shot models on out-of-distribution dynamics by studying a dataset in which chaotic systems continuously lose energy due to processes like dissipation. This makes the dynamics non-stationary and the forecasts need to describe a different dynamical regime than the context. We found that, across all 135 systems, the forecast accuracy degrades as the degree of non-stationarity is increased (Figure 11 and Appendix E). This matches our expectation context matching is a key strategy used by Chronos.

---

> ### Author Response · Authors · 2024-11-25
>
> Hello, since the review period is ending soon, we wanted to check back to see if the referee has any additional questions or comments that we can address.
>
> You can see a summary of our major changes above, but the main new experiments relevant to this review are: (1) a new figure and appendix on zero-shot forecasting of real-world experimental chaotic systems, (2) two new figures and an appendix on sensitivity to initial conditions, and (3) a new figure and appendix on out-of-distribution generalization.
>
> Thanks very much for your review and feedback on our paper! We think the new version has improved a lot.

---

> > ### Comment · Reviewer_yZ6s · 2024-11-30
> >
> > Thank you for the clarifications and new experiments. I've raised my score.

---

> > > ### Author Response · Authors · 2024-11-30
> > > **Thank you**
> > >
> > > Thank you very much for your review and for raising your score, we very much appreciate it.

---

### Official Review · Reviewer_1G8R · 2024-11-01

**Soundness:** 3
**Presentation:** 3
**Contribution:** 3
**Rating:** 6
**Confidence:** 4

**Summary:**

The article explores the application of foundation models, particularly zero-shot learning, to forecast chaotic systems. Using the foundation model Chronos, the study investigates whether these models can generate accurate short-term forecasts and capture the long-term statistical properties of chaotic attractors, which are characteristic of chaotic systems. Through an extensive evaluation on 135 chaotic systems, the authors found that, even without specific training on chaotic dynamics, Chronos provides competitive forecasts, preserving geometric and statistical attributes of chaotic attractors over the long term. The study presents these findings as a benchmark for the feasibility and limitations of using foundation models in scientific machine learning, especially in physics-informed tasks.

**Strengths:**

1. Extensive Benchmarking: The authors conduct a thorough and extensive evaluation, with a dataset that includes 135 chaotic systems, providing statistically robust results.
2. Foundation Model Scalability: By demonstrating the scalability of Chronos across varying chaotic systems, the study establishes the potential of foundation models for scientific machine learning in challenging domains.
3. Zero-Shot Learning Feasibility: The findings validate that foundation models can make meaningful forecasts in chaotic systems with minimal domain-specific adjustments, a promising step for generalizing AI across diverse scientific domains.
4. Insight into Long-Term Statistical Consistency: The study’s focus on long-term attractor behavior offers a novel way to assess model performance, moving beyond conventional short-term forecast accuracy.

**Weaknesses:**

1. The article mentions that Chronos’s forecast accuracy is sensitive to initial conditions, but the specifics of this dependency aren’t deeply explored. Could the authors provide a more systematic investigation into how initial condition variability impacts model robustness? For example, have you considered evaluating Chronos’s performance across a range of initial conditions sampled from different regions of the attractor, particularly comparing accuracy in central versus peripheral areas? Additionally, could you quantify how forecast accuracy shifts as the initial conditions deviate from those in the training data? Such analyses would offer valuable insights into Chronos’s generalization capabilities and help identify any specific sensitivities related to initial condition variability.
2. Positional Encoding and Temporal Structure: Given the chaotic nature of the systems studied, the choice of positional encodings (e.g., rotary embeddings) could significantly impact model performance, especially in maintaining temporal coherence over long horizons. However, the article lacks an in-depth discussion or ablation study regarding the choice of these encodings.
3. Limited Explanation of Performance in Extreme Cases: The authors mention that larger models tend to perform better, yet details on why model size improves stability in chaotic systems are sparse. A theoretical or empirical justification of this scaling behavior, such as through model capacity for capturing non-linear dynamics, would improve understanding.
4. Scalability of Model Parameters and Computational Requirements: Although Chronos shows promise in forecasting chaotic systems, the article could discuss practical limitations regarding computational costs, especially for larger model sizes, and the trade-offs compared to specialized models like NBEATS or TiDE.

**Questions:**

1. Impact of Initial Conditions: Can the authors elaborate on how initial conditions affect Chronos’s zero-shot forecasting stability? Is there a significant dependency on starting points, especially when far from the attractor’s typical state?
2. Effectiveness of Positional Encodings: How critical is the choice of positional encodings for Chronos when applied to chaotic systems? Have other encoding methods been tested, and if so, how did they affect both short-term accuracy and long-term attractor preservation?
3. Model Complexity and Forecasting Horizon: Given that larger models perform better in long-term forecasting, could the authors provide insights into the mechanisms by which increased model size aids in managing chaotic variability?
4. Comparison with In-Weights Fine-Tuning: Since the study focuses on zero-shot learning, have the authors considered comparing this approach with in-weights fine-tuning on a few chaotic trajectories to assess the performance improvement?

---

> ### Author Response · Authors · 2024-11-13
> **Clarifying questions at start of discussion period**
>
> **[Removed, please see our detailed replies below]**

---

> ### Author Response · Authors · 2024-11-21
> **Detailed Reply (1/2)**
>
> Thank you very much for your positive review and comments! We very much appreciate your advice and interest in the paper. We have made several changes (listed below) that may be of interest; please let me know if you have any additional questions or feedback.
>
> **[INITIAL CONDITIONS]**
>
> *See revised text Fig. 5, Fig. 9, Fig. 11 and new Appendices C and E.*
>
> We performed several analyses to answer the question: how does initial condition variability impact zero-shot forecasting accuracy? In general, we confirmed that there is significant variability of forecast accuracy depending on the initial condition (spanning over two orders of magnitude of the VPT, see Figure 9). Moreover, we found a weak but robust correlation between the natural measure on the attractor and forecast accuracy (Appendix C and Figure 10). This is consistent with the idea that zero-shot forecast models perform better at forecasting denser, more common regions of the attractor, because those points appear more often in the context. Conversely, rarer points (i.e., extremal points closer to out-of-distribution dynamics relative to the context points) lead to worse forecasts.
>
> Most interestingly, we found that there is a much stronger correlation between context overlap and forecast accuracy (Figure 6). This analysis was motivated by the following observation: If you look at the first trajectory in Figure 5, you will see that Chronos seems to be copying and pasting part of the context trajectory in the successful forecast. This strategy (looking for snippets in the context trajectory that most closely resemble the history immediately preceding the prediction and simply repeating that pattern) is extremely simple but can actually be very effective. You can see Chronos trying to do the same thing for the second trajectory, only that this time it cannot find a very good repeating pattern in the context trajectory, so the prediction is worse. From the perspective of Takens’ embedding theorem, this strategy aims to find the closest context data point to the initial condition in the delay embedding space and repeating the context trajectory from that point. We hypothesize that this is one of the strategies used by Chronos for zero-shot forecasting, which led us to compute the minimum distance between the context points and the initial condition in the delay embedding space. We found that this minimum distance is a good predictor of the forecast accuracy (Fig. 5), which explains the observed variability of forecast performance on initial conditions and led credence to the idea of context matching as a mechanism for zero-shot forecasting.
>
> **[POSITIONAL ENCODING]**
>
> *See revised text Fig. 6A*
>
> We perfomed new experiments by randomly shuffling all k-grams of timepoints in the model's context. We kept the last k context timepoints the same as the original training dataset, but made sure the penulative k-gram differed.
> As a baseline, we also directly performed zero-shot forecasts using only the last k context timepoints.
> We found that random k-gram shuffles of longer contexts provide better forecasts than shorter context baselines, suggesting that Chronos can utilize the context data even without positional information (Figure 6 and Section 5.3). We believe this is because earlier context points provide information about the natural measure of the attractor, as well as conditional probabilities of successive timepoints.
>
> While early points in a long context are decorrelated with the predictions, they are drawn from the same underlying distribution, and so longer contexts provide information about the distribution of attractor states regardless of positional encoding. This is important because Chronos (which tokenizes time series, and then uses a T5 encoder-decoder model language model) lacks an explicit positional encoding mechanism.
>
> **[MODEL SIZE]**
>
> *See revised text Fig. 6*
>
> We have added new experiments analyzing the scaling laws for the performance of Chronos with model size and context length. Forecasts improve with larger model size, a phenomenon that we attribute to in-context learning (ICL), which also underlies scaling of language model performance. In Figure 6A, we ablate ICL by deliberately shuffling the context and seeing how the performance of Chronos with a context of $512$ tokens changes. We then re-introduce structure by preserving all length-2, length-3, etc sequences, and we observe that the performance of Chronos recovers. This suggests that ICL in Chronos manifests as learning conditional dependencies among tokens, which matches findings for language models. To further confirm this effect, our new experiments on initial conditions (Fig. 5, Fig. 9, Fig. 11 and new Appendix C) show that perturbations that change the context similarity with the test setting also reduce zero-shot forecast performance.

---

> ### Author Response · Authors · 2024-11-21
> **Detailed Reply (2/2)**
>
> **[FINE-TUNING]**
>
> We did try this, but the results were inconclusive enough that we decided our setting wasn't a fair experiment. We tried taking the entire set of trajectories we used for this paper and fine-tuning Chronos-base (for which training scripts are available). However, we weren't able to appreciably move the validation scores, and the score actually got worse on the original Chronos dataset. This suggests to us that the properties of the dynamical systems dataset significantly differs from the datasets on which Chronos was trained. I expect that fine-tuning Chronos for this task will require either a LoRA-like adaptation trick, or substantially more data and compute. For now, we've added some text to the discussion mentioning that we tried this, but we think that it needs to be addressed in future work as a standalone project.
>
> **[OUT-OF-DISTRIBUTION GENERALIZATION]**
>
> In new experiments (Fig. 11, Appendix E) we probe the performance of zero-shot models on out-of-distribution dynamics by applying exponential modulations to the chaotic time series, which mimics physical systems losing energy due to processes like dissipation. This makes the dynamics non-stationary and the forecasts need to describe a different dynamical regime than the context. We found that, across all 135 systems, the forecast accuracy degrades as the degree of non-stationarity is increased (Figure 11 and Appendix E). This matches out expectation given that we found context matching to be a key strategy used by Chronos.

---

> ### Author Response · Authors · 2024-11-25
>
> Thanks again for reviewing our paper. As the review period winds down, we wanted to circle back to see if the referee has had the chance to consider our revision, and if they have any further questions.
>
> To recap the changes related to this review, we added new experiments (with figures and appendices) in response to the referee’s requests regarding: (1) initial conditions sensitivity,  (2) ablation of positional encodings, and (3) scaling laws for model complexity and in-context learning, as well as new work showing a forecasting mechanism via context parroting.

---

> > ### Comment · Reviewer_1G8R · 2024-11-25
> > **More clarification needed**
> >
> > **Response to [FINE-TUNING]:**
> > Thank you for providing details on your experiments with fine-tuning Chronos-base. Your observation that fine-tuning degraded performance on the original Chronos dataset is insightful and highlights potential limitations of transferring Chronos to fundamentally different datasets. The mention of needing a LoRA-like adaptation or more extensive data/computational resources is intriguing.
> >
> > Could you elaborate on the challenges you faced during fine-tuning? For instance:
> > 1. Did you observe overfitting, instability, or specific patterns in the validation loss that could point to optimization issues?
> > 2. Was the degradation in performance consistent across all chaotic systems, or did certain systems benefit from fine-tuning?
> > 3. Have you considered alternative regularization techniques or domain-specific pretraining steps to bridge the gap between the original and chaotic datasets?
> >
> > Adding more details about these challenges could clarify the potential research directions for future studies on fine-tuning foundation models for chaotic systems.
> >
> > ---
> >
> > **Response to [OUT-OF-DISTRIBUTION GENERALIZATION]:**
> > Your inclusion of new experiments (Figure 11, Appendix E) is an excellent addition to the study, as it provides empirical evidence for Chronos’s behavior under non-stationary dynamics. The observation that forecast accuracy degrades with increasing non-stationarity aligns with expectations and further demonstrates the importance of context matching in Chronos’s strategy.
> >
> > Could you provide additional insights into the following:
> > 1. Did you explore whether certain types of chaotic systems (e.g., high-dimensional attractors versus lower-dimensional ones) exhibit greater resilience to non-stationarity?
> > 2. Are there specific modifications to the zero-shot framework or context selection strategies that could mitigate performance degradation under non-stationary conditions?
> > 3. How does Chronos perform compared to fine-tuned or domain-specific models on these out-of-distribution tasks?
> >
> > Your results open up interesting questions about how well foundation models can generalize to new dynamical regimes, and addressing these could inspire further developments in the field.

---

> > > ### Author Response · Authors · 2024-11-27
> > > **Answers to clarifying questions; new revision posted.**
> > >
> > > Thank you very much for your additional comments and response, and for affirming the insight and general interest of our paper. We agree, and have incorporated responses to all your question within the revised PDF we just posted:
> > > + Responses to *[FINE-TUNING]* are found in new *Appendix F.2*
> > > + Responses to *[OUT-OF-DISTRIBUTION GENERALIZATION]* have been incorporated into Section 5.3 and Appendix E and Fig. 11.
> > >
> > > Below, we summarize the information we added in the revision:
> > >
> > > **[FINE-TUNING]:**
> > > 1. We observe a shallow plateau in the validation loss. The model initially seems to improve, but it stops quickly, and we see no qualitative change in the zero-shot forecasts. We observe this even in single systems, like the Lorenz attractor, but it persists even when we use all of the data used elsewhere in the study. We therefore conclude that directly fine-tuning a model of the scale of Chronos requires a substantially larger dataset than the one used for the zero-shot and existing baseline fully-trained models.
> > >
> > > 2. We do not observe a consistent relationship between system properties and training instability. Our hypothesis is that we are in a training regime where the model can’t adapt to properties of individual time series. One reason for this is that our chaotic systems time series have properties that strongly differ from Chronos’s training corpus (smoothness, strong channel coupling, etc), and so during training the model generically sees new data as strongly out-of-distribution. We think retraining or even architectural modifications will be needed to get around this.
> > >
> > > 3. We agree that domain-specific adaptations could mitigate this. In particular, Chronos’s tokenizer and the internal encoder-decoder model could be replaced with models that account for strong channel coupling observed in chaotic systems, which differs from many other multivariate time series like those used to train Patch-TST and other leading models. We also think we could use a training schedule that mixes dynamical systems data in with the original data on which Chronos-base was trained. As a fallback, recently versions of low-rank adaptation have been proposed for forecast models. However, we think is it more appropriate to do this in future work, because there are several additional considerations regarding how training data is curated and how the model encounters different dynamical regimes, which require significant engineering effort and potentially full retraining with a novel architecture.
> > >
> > > **[OUT-OF-DISTRIBUTION GENERALIZATION]**
> > > 1. Please see newly-revised Appendices D and E. The dimensionality of a system’s dynamics doesn’t strongly correlate, since a highly-nonlinear low-dimensional system may be harder to forecast than a weakly-nonlinear high-dimensional system, and concepts like Koopman operator theory suggest a certain tradeoff between these regimes. However, we do observe the same dependence on chaoticity as we observe on in-distribution zero-shot forecasts (Fig. 14). Generally, the degree to which the zero-shot model sees a time series as OOD depends on a combination of the context points’ density on the attractor (the rarity of individual timepoints, pairs of timepoints, etc) and the overall degree of nonstationarity (e.g. distribution shift). Thus, because attractor density weakly correlates with Lyapunov exponent and fractal dimension (two dynamical invariants that also affect density variation), we believe that correlations of OOD performance with invariant dynamical properties are secondary effects of variation in attractor density.
> > >
> > > 2. Please see newly-revised Appendix E and Section 5.3. The context-parroting we report in Section 5.3 is a known behavior of large language models, and can be adapted to out-of-distribution setting with similar strategies as those used in LLMs. For example, Chronos currently does not use explicit positional encoding (it feeds tokens into a T5 encoder-decoder model), and so the current model does not explicitly include time information in the context. Adding rotary encoding would allow Chronos to anticipate simple nonstationarities like monotonic trends, which smaller fully-trained models easily handle. Additionally, because zero-shot forecast models generate probabilistic forecasts, their sampling scheme can be manipulated towards less “equilibrium” (rarer) trajectories, which would favor irreversible and thus non-stationary trajectories.
> > >
> > > 3. Please see revised Appendix E. Fully-trained small models like NBEATS perform better on this task. Many fine-tuned models are designed to handle predominant “trend” components, such as the monotonic decrease that we study in Appendix E. As a result, these models can fit a trend during training and easily continue it during context. In order to observe this phenomenon in a zero-shot setting like Chronos, we believe it would be necessary to make the modifications we list above, such as using positional embeddings to give explicit time information.

---

> > > > ### Comment · Reviewer_1G8R · 2024-11-27
> > > > **Final Feedback**
> > > >
> > > > Thank you for your detailed and thorough responses to my questions and for incorporating additional insights into the revised paper. I appreciate the effort to address my concerns both in the main text and through the newly added appendices.
> > > >
> > > > ### Response to [FINE-TUNING]
> > > > I appreciate the clarification on the challenges encountered during fine-tuning Chronos, particularly regarding the plateau in validation loss and the observations related to out-of-distribution data properties. Your hypothesis about the model’s inability to adapt to individual chaotic systems due to differences in training data characteristics is insightful. I agree that retraining or architectural modifications, such as adapting the tokenizer and encoder-decoder model to account for strong channel coupling, are promising directions for future work. The proposed idea of mixing dynamical systems data with the original Chronos training corpus is particularly compelling.
> > > >
> > > > The discussion on low-rank adaptation strategies and the acknowledgment of the significant engineering effort required for effective fine-tuning further demonstrate a nuanced understanding of the problem. I find your decision to defer this exploration to future work reasonable and well-justified.
> > > >
> > > > ### Response to [OUT-OF-DISTRIBUTION GENERALIZATION]
> > > > The additional experiments and analyses in Appendices D and E are excellent additions. The explanation regarding attractor density, its weak correlation with Lyapunov exponent and fractal dimension, and the resulting impact on out-of-distribution generalization offers a deeper understanding of Chronos’s behavior. I found your insights on how attractor density influences the model’s performance, combined with the notion of context density and nonstationarity, very informative.
> > > >
> > > > The proposed modifications, such as adding rotary positional embeddings to account for temporal information and adapting context-parroting strategies from large language models, appear promising for enhancing Chronos’s ability to handle nonstationary dynamics. These ideas not only address my initial concerns but also outline clear future directions for improving zero-shot forecast models like Chronos.
> > > >
> > > > ### General Feedback
> > > > Your revisions provide a well-rounded discussion of the limitations and potential enhancements for Chronos. The added context on how zero-shot models can be adapted to better handle nonstationary regimes is especially valuable. Additionally, the acknowledgment that smaller, fully-trained models like NBEATS currently outperform Chronos on specific tasks highlights the paper’s balanced approach to evaluating the state of the field.
> > > >
> > > > Overall, the revised paper significantly improves clarity and depth, and I believe your additions will contribute meaningfully to advancing research in this area. Thank you again for your thoughtful and comprehensive responses.

---

> > > > > ### Author Response · Authors · 2024-12-02
> > > > >
> > > > > Thank you for your reply! Given that the referee appears satisfied with our response and is overall positive on the paper, we ask if you will consider updating your score in the original review, to reflect your more positive evaluation.
> > > > > Thank you for considering our request.

---

### Author Response · Authors · 2024-11-21
**Summary of Major Changes**

Thank you very much to the reviewers for their comments and suggestions! We have posted a substantially revised manuscript addressing all comments and requests for additional experiments with **6 new Figures and 4 new Appendices.** We believe the manuscript has really improved as a result.

We have responded to each referee individually below; here we summarize the major changes:

**[EXPERIMENTS]**
- Two new figures and a new appendix systematically showing how initial condition variability impacts zero-shot forecasting accuracy. This offers new insights into the inner workings of time-series foundation models (*1G8R* & *yZ6s*)
- A new figure and experiments confirming that Chronos performs in-context learning, even with shuffled positional encoding (*1G8R*)
- New experiments and a new Figure with forecasts of a real-world chaotic system (video of an oscillating double pendulum) (*yZ6s*)
- New experiments demonstrating that zero-shot models can handle out-of-distribution tasks (*1G8R* & *yZ6s*)

**[EXPOSITION]**
- Provided more details on the LSTM and transformer models (*MDoC*)
- Further optimized the presentation of the paper with more descriptive subsection titles and clearer segues (*MDoC*)
- Modified the title to be more concise: "Zero-shot forecasting of chaotic systems"

Please let us know if you have any additional comments or requests.

---

### Meta-Review · Area_Chair_im65 · 2024-12-15

**Metareview:**

The paper explores zero-shot learning capabilities of foundation models, specifically Chronos, for forecasting chaotic systems. It demonstrates that Chronos can perform competitively against custom-trained models across 135 chaotic systems, capturing both short-term accuracy and long-term statistical properties of chaotic attractors. Strengths include its extensive benchmarking, validation of zero-shot learning in chaotic systems, and insights into foundation models’ scalability and in-context learning mechanisms. Weaknesses noted by reviewers include sensitivity to initial conditions, limited exploration of real-world chaotic systems, and underdeveloped discussions on scalability and computational trade-offs. Missing elements include further baselines like Mamba and more real-world applications. While the study highlights the potential of zero-shot models in chaotic system forecasting, limitations in robustness and the computational demands of foundation models suggest incremental but promising advancements. The reviewers’ consensus to accept the paper reflects its significant contributions to applying zero-shot learning in scientific machine learning, despite areas for future work.

**Additional Comments On Reviewer Discussion:**

During the rebuttal period, reviewers raised concerns about Chronos’s sensitivity to initial conditions, the impact of positional encodings on performance, scalability, computational efficiency, and the limited exploration of real-world chaotic systems. The authors addressed these by adding new experiments and analyses, including studies on initial condition variability, context matching, and real-world chaotic system forecasting (e.g., an oscillating double pendulum). They clarified model details, provided insights into scaling laws and in-context learning, and discussed the challenges of fine-tuning on chaotic datasets. While reviewers appreciated the detailed responses and new experiments, some concerns about computational costs and broader applicability remained. The revisions significantly enhanced the manuscript’s clarity and depth, leading to an overall consensus for acceptance based on its strong contributions to understanding zero-shot learning in chaotic systems.

---

### Decision · Program_Chairs · 2025-01-22

Accept (Poster)